# Deeper with Riemannian Geometry: Overcoming Oversmoothing and Oversquashing for Graph Foundation Models

**Li Sun**[*]
Beijing University of Posts and Telecommunications
Beijing 100876, China
lsun@bupt.edu.cn

**Zhenhao Huang**
North China Electric Power University
Beijing 102206, China
huangzhenhao@ncepu.edu.cn

**Ming Zhang**
North China Electric Power University
Beijing 102206, China
zhangming@ncepu.edu.cn

**Philip S. Yu**
University of Illinois Chicago
IL 60607, USA
psyu@uic.edu

## Abstract

Message Passing Neural Networks (MPNNs) are the building block of graph foundation models, but fundamentally suffer from oversmoothing and oversquashing. There has recently been a surge of interest in fixing both issues. Existing efforts primarily adopt **global** approaches, which may be beneficial in some regions but detrimental in others, ultimately leading to the suboptimal expressiveness. In this paper, we begin by revisiting oversquashing through a global measure – spectral gap $\lambda$ – and prove that the increase of $\lambda$ leads to gradient vanishing with respect to the input features, thereby undermining the effectiveness of message passing. Motivated by such theoretical insights, we propose a **local** approach that adaptively adjusts message passing based on local structures. To achieve this, we connect local Riemannian geometry with MPNNs, and establish a novel nonhomogeneous boundary condition to address both oversquashing and oversmoothing. Building on the Robin condition, we design a GBN network with local bottleneck adjustment, coupled with theoretical guarantees. Extensive experiments on homophilic and heterophilic graphs show the expressiveness of GBN. Furthermore, GBN does not exhibit performance degradation even when the network depth exceeds 256 layers.

## 1 Introduction

Recently, graph foundation models have attracted growing research interests [77; 12; 68; 61], offering the pre-trainable and general-purpose deep architectures based on Message-Passing Neural Network (MPNN). Against this backdrop, we revisit the fundamental issues of the core building block – the MPNN itself: *Oversmoothing* [35; 40], which prevents MPNN from going deeper as other typical neural architectures, and *Oversquashing* [1; 64], which refers to the difficulty in propagating information from distant nodes and capturing the long-range dependencies.

In the literature, there has recently been a surge of interest in jointly addressing oversquashing and oversmoothing. The former is caused by the topological "bottlenecks" in the graph structure [64], while the latter can also be related to the structure [11; 44]. Accordingly, a primary solution is graph rewriting, which conducts the structural refinement guided by Ricci curvature [39; 38; 20] or spectral

---

[*]Corresponding Author: Li Sun, lsun@bupt.edu.cn

39th Conference on Neural Information Processing Systems (NeurIPS 2025).

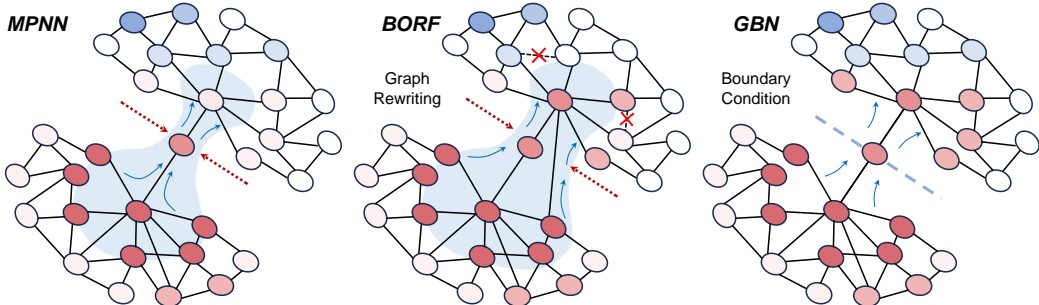

Figure 1: **Illustration of different solutions against oversquashing.** (A) In typical MPNNs, messages are "squashed" at the bottleneck of the graph structure. (B) In graph rewriting (e.g., BORF [39]), oversquashing at the bottleneck is mitigated by adding edges, but the removal of edges (in light of oversmoothing) hinders the message-passing in other regions. (C) In the proposed GBN, an adaptive adjustment based on the local Riemannian geometry is introduced to message passing in each region to enhance expressiveness.

analysis [4]. The underlying rationale lies in the global metric of spectral gap (i.e., the first non-zero eigenvalue of graph Laplacian $\lambda$), which controls the size of the bottleneck [6]. On the one hand, graph rewriting introduces significant computational overhead. On the other hand, it undergoes a carefully designed edge adding-deleting procedure [2], altering the original graph structure. So far, existing efforts, including the recent advance [27], adopt the **global** approach to fix MPNNs.

In this paper, we begin by rethinking a significant question: *is it necessary to increase the spectral gap $\lambda$ through a **global** approach?* Intuitively, a global approach may benefit some regions but be detrimental to others. Theoretically, a larger $\lambda$ accelerates the oversmoothing process, as measured by Dirichlet energy [35; 40]. Regarding oversquashing, we develop a unified interpretation of typical MPNNs as neural solvers of the heat equation on Riemannian manifold, which reveals the negative impact of increasing $\lambda$ through the eigenvalue decomposition of Riemannian heat kernel. Specifically, increasing $\lambda$ tends to cause gradient vanishing (Theorem 4.2), thereby limiting the model expressiveness.

Motivated by the theoretical insights above, we aim to address both issues of MPNN through a **local** approach. Instead of the structural refinement, we focus on refining the message passing paradigm, and the key innovation lies in the local bottleneck adjustment according to the subgraph structure. To achieve this, we explore the functional space over subgraphs through its continuous counterpart – the local Riemannian geometry over submanifolds. Concretely, we establish a novel **nonhomogeneous boundary condition** that is rigorously proven to mitigate both oversquashing and oversmoothing, thereby allowing for MPNNs to go deeper. Consequently, we design a simple yet effective Graph Boundary conditioned message passing Neural network (**GBN**) to implement the solver subject to the nonhomogeneous boundary condition. GBN unifies the updating rules for boundary and interior nodes with satisfactory scalability, accompanied with the favorable theoretical guarantees.

**Contributions.** In summary, our key contributions are three-fold: **A. Insight.** On oversquashing, we reveal a limitation of the global approach via the spectral gap, motivating a local approach where message passing is adaptively adjusted at each node. **B. Theory.** We introduce the concept of Riemannian boundary condition to address oversquashing — to the best of our knowledge, for the first time — and establish a nonhomogeneous boundary condition that also enables deeper MPNNs. **C. Methodology.** Grounded in the Robin condition, we propose a new MPNN named GBN with theoretical guarantees. Extensive experiments on both homophilic and heterophilic graphs demonstrate the expressive power of GBN.

## 2   Related Work

**Oversmoothing.** Previous solutions to mitigate oversmoothing can be roughly categorized into four groups: ① Residual connections [72; 36; 74], particularly initial residual connections [13], and

---

[2]Graph rewriting methods typically add edges to enlarge bottlenecks, while removing some dense connections in light of oversmoothing.

② Normalization layers [25; 78; 79] mitigate oversmoothing through the lens of gradient vanishing. Recently, [47] demystifies how they affect the expressive power of MPNNs, while [3] develops a principled explanation via gradient vanishing, and bridges MPNNs and recurrent networks. ③ Gating mechanism [8; 75; 76; 46] modulates gradient updates in the layers, and allows for different rates of message passing. ④ Graph sparsification [11; 44] combats this issue from the structural perspective, given that some subgraphs, such as cliques, increase the risk of oversmoothing [39]. A theoretical review is provided in [31]. ⑤ [32] presents a novel idea to transform the global spectral distribution in each layer of MPNN, while we focus on local adjustment of MPNN via boundary conditions. Different from designing additional modules or structural refinement, we focus on refining MPNN mechanism, not susceptible to oversmoothing.

**Oversquashing.** ① It strongly depends on the graph structure, and thus a natural solution is graph rewriting, which typically adds edges to enlarge the topological bottlenecks, guided by Ricci curvature [64; 2; 18] or spectral analysis [30]. Also, graph transformers can be regarded as their generic form with learnable attentional weights [70; 21; 49]. [7] understands graph rewriting from the lens of effective resistance. Recently, it has been shown that incorporating virtual nodes is effective [45; 50]. [23] provides the theoretical aspects of oversquashing on depth, width, and topology of graphs. ② Recent studies jointly address oversmoothing via global treatments. In the line of graph rewriting, [39; 38; 20] introduce edge addition-deletion algorithms in light of oversmoothing; [28] increases the spectral gap via edge deletion; [4] conducts the feature rewriting with Delaunay triangulation. Recently, [27] considers the polynomial bases for the graph Fourier transform, while [42] designs an auxiliary module to prevent heterophily mixing. To combat over-squashing, [16] presents a novel approach to select nodes with high centrality to expand in width to encapsulate the growing influx of signals from distant nodes. In contrast to the previous global treatments, we introduce the local adjustment to message-passing paradigm with nonhomogeneous boundary conditions.

**Riemannian Geometry & MPNNs.** Riemannian geometry has been introduced to learning on graphs, e.g., node clustering [58; 57; 55], node classification [53; 59; 60; 22], structural learning [54], modeling dynamics [56; 52; 51; 62] and generative models [14; 69; 63], and shown potential to graph foundation models [61]. In this context, efforts are made to extend Euclidean MPNNs to geodesically complete manifolds, (e.g., hyperbolic spaces [37; 10; 51], stereographic models [5], product spaces [53] and pseudo Riemannian manifolds [34; 71]), seeking embeddings from graphs to specific manifolds. Recently, the discrete versions of Ricci curvature, the concept describing the structural connectivity [65], have been introduced to address oversmoothing and oversquashing [39; 38; 20; 64]. In contrast, we consider the functional spaces over Riemannian manifold.

## 3 Preliminaries

We review the heat flow on Riemannian manifold, the foundation of our analysis, and describe the oversquashing and oversmoothing in MPNNs. The notation table is provided at Appendix A.

**Riemannian Manifold & Heat Flow.** In Riemannian geometry, a graph is regarded as the discrete analogy to certain Riemannian manifold [65]. $(\mathcal{M}, g)$ denotes a $n$-dimensional manifold endowed with the Riemannian metric $g$.[3] The volume form is written as $\mathrm{vol}_{\mathcal{M}} = \sqrt{|\det g|} dx^1 \wedge \cdots \wedge dx^n$, where $\boldsymbol{x} \in \mathbb{R}^n$ denotes local coordinates and $x$ is the element of $\boldsymbol{x}$. A Hilbert (function) space $L^2(\mathcal{M}) = \{f \in C^\infty(\mathcal{M}) | \int_{\mathcal{M}} f \, \mathrm{vol}_{\mathcal{M}}\}$ consists of smooth $L^2$ integrable functions on $\mathcal{M}$. Let $\Delta_g f = \mathrm{div}(\mathrm{grad}\, f)$ be the Laplace-Beltrami operator on $\mathcal{M}$ *w.r.t.* $g$. $\Delta_g$ on a closed manifold has discrete eigenvalues $\lambda$, and its eigenfunctions $\psi$ form a basis of $L^2(\mathcal{M})$. A homogeneous heat flow on $\mathcal{M}$ is described as the following differential equation

$$\partial_t u(t, \boldsymbol{x}) = \Delta_g u(t, \boldsymbol{x}), \ \ t > 0 \tag{1}$$

with the initial condition $u(0, \boldsymbol{x}) = \phi(\boldsymbol{x})$ at $t = 0$. The solution is induced by the heat kernel $H(t, \boldsymbol{x}, \boldsymbol{y}) : \mathcal{M} \times \mathcal{M} \to \mathbb{R}$ so that $u(t, \boldsymbol{x}) = e^{t\Delta_g} f := \int_{\mathcal{M}} H(t, \boldsymbol{x}, \boldsymbol{y}) f(\boldsymbol{y}) \, \mathrm{vol}_{\mathcal{M}}(\boldsymbol{y})$.

**MPNN, Oversquashing & Oversmoothing.** Let $\mathcal{G} = (\mathcal{V}, \mathcal{E})$ be an undirected and unweighted graph on the node set $\mathcal{V}$ and edge set $\mathcal{E} \subseteq \mathcal{V} \times \mathcal{V}$. The nodes are associated with a feature matrix

---

[3]Throughout this paper, the terminology of manifold $\mathcal{M}$ refers to a closed Riemannian manifold (compact and without boundary) unless otherwise specified.

$\mathbf{X} \in \mathbb{R}^{N \times d}$, where $|\mathcal{V}| = N$ is the number of nodes, and $d$ denotes the dimension of input features. The structural information is encoded in a binary adjacency matrix $\mathbf{A}$ and, accordingly, the normalized Laplacian of $\mathcal{G}$ is written as $\mathbf{L} = \mathbf{I} - \mathbf{D}^{-\frac{1}{2}} \mathbf{A} \mathbf{D}^{-\frac{1}{2}}$, where $\mathbf{D}$ is the diagonal degree matrix. A Message Passing Neural Network (MPNN) recursively conducts neighborhood aggregation by $K$ layers, and node representation $\boldsymbol{x}_v$ of the $k$-th layer is updated with the following function,

$$\boldsymbol{x}_u^{(k)} = \varphi^{(k)}(\boldsymbol{x}_u^{(k-1)}, \mathrm{Agg}(\{\boldsymbol{x}_v^{(k-1)} : (u,v) \in \mathcal{E}\})), \tag{2}$$

for $k = \{1, \dots, K\}$, where Agg is a permutation invariant aggregation function, and $\varphi^{(k)}$ denotes the update function. MPNNs suffer from *Oversquashing* [1; 64], the compression of messages from exponentially expanding receptive fields into fixed-size vectors. It is typically characterized by the Jacobian $\|\partial \boldsymbol{x}_u^{(K)} / \partial \boldsymbol{x}_v^{(0)}\|$, and the exponential decay of derivative with respect to the hop distance implies the occurrence of oversquashing. *Oversmoothing* [35; 40] refers to the phenomenon that node representations become indistinguishable to each other *exponentially* as the number of layers increases. **In this paper, we aim to address both issues of MPNN through a novel local approach that leverages the local Riemannian geometry.**

## 4 Theoretical Insights on Oversquashing

We first develop a unified interpretation of MPNNs from the lens of Riemannian geometry. Then, regarding oversquashing, we demonstrate the negative impact of increasing the spectral gap $\lambda$, which motivates us to reconsider: *is it necessary to increase $\lambda$ through a **global** approach?*

**A unified formalism.**    In order to provide a principled analysis, we construct a unified formalism in Riemannian geometry as its notions align with those in spectral graph theory [17, Chap. 10]. To be specific, we first construct the following layer-wise heat equation analogous to Eq. (1).

$$\begin{cases} \partial_t u_k(t, \boldsymbol{x}) = \Delta_g u_k(t, \boldsymbol{x}), & k = 1, \cdots, K \\ u_k(t_k, \boldsymbol{x}) = \phi_k(\boldsymbol{x}), & k = 1, \cdots, K \end{cases} \tag{3}$$

with the initial condition $\phi_1(\boldsymbol{x}) = \phi(x)$ of input feature, where $0 = t_1 < t_2 < \dots < t_K < t_{K+1} = T$ and $t_{k+1} - t_k = \Delta t$ for each layer $k = 1, \dots, K$. Second, the solution is derived as follows,

$$\begin{cases} u_k(t, \boldsymbol{x}) = \int_{\mathcal{M}} H(t - t_k, \boldsymbol{x}, \boldsymbol{y}) \phi_k(\boldsymbol{y}) \mathrm{vol}(\boldsymbol{y}), & k = 1, \cdots, K, \\ \phi_k(\boldsymbol{x}) = G_{\theta_k}[u_{k-1}(t_k, \boldsymbol{x})], & k = 2, \cdots, K, \end{cases} \tag{4}$$

where $G_{\theta_k}$ is a learnable neural net with linear transformations. Third, with input representation $\phi_k(v_j)$, the output of $v_i$ at the $k$-th layer $u_k(t_k, v_i) = \sum_{j \in \mathcal{V}} H(\Delta t, v_i, v_j) \phi_k(v_j)$ is given by the **heat kernel** $H(t, v_i, v_j) := [\mathbf{H}_t]_{ij}$ and $\mathbf{H}_t = e^{-t\mathbf{L}}$ [17]. According to the matrix exponential $e^{-t\mathbf{L}} = \mathbf{I} + \sum_{k=1}^{\infty} \frac{1}{k!} (-t\mathbf{L})^k$, if we take a small time step $\Delta t \to 0$, the first-order approximation is given by $e^{-\Delta t \mathbf{L}} \sim \mathbf{I} - \Delta t \mathbf{L}$. Therefore, we have the updating rule in matrix form,

$$\mathbf{U}_k = \mathbf{H}_{\Delta t} \boldsymbol{\Phi}_k = (\mathbf{I} - \Delta t \mathbf{L}) \boldsymbol{\Phi}_k, \quad \boldsymbol{\Phi}_k = G_{\theta_k}(\mathbf{U}_{k-1}). \tag{5}$$

For example, GCN [33] is an instance of Eq. (5) with $\Delta t = 1$. That is, *typical MPNNs in Eq. (2) can be interpreted as a neural solver to the heat equation over a Riemannian manifold*, offering the theoretical foundation of this paper. The following elaboration considers the scalar-valued function for clarity.

**On oversquashing: revisiting the spectral gap**    Arranging the eigenvalues of the Laplacian of manifold heat kernel in ascending order, the first non-zero eigenvalue is referred to as *spectral gap $\lambda$*. From the topological perspective, we connect the value of $\lambda$ to the topological "bottleneck" from Riemannian geometry, and the connection is established via Cheeger's constant. Concretely, for a given manifold $\mathcal{M}$, the *Cheeger's constant* $h_{\mathcal{M}} = \inf_S \frac{\mathrm{Area}(\mathcal{S})}{\min\{\mathrm{vol}(\mathcal{A}), \mathrm{vol}(\mathcal{B})\}}$ is related to the area of the bottleneck's cross-section dividing the $\mathcal{M}$ into two disjoint submanifolds $\mathcal{A}$ and $\mathcal{B}$. [4]

**Theorem 4.1** (**Cheeger and Buser's Inequality [9]**). *If the Ricci curvature of manifold $\mathcal{M}$ is bounded by $-(n-1)\xi^2$, and $\xi \geq 0$, then $\frac{h_{\mathcal{M}}^2}{4} \leq \lambda_1 < C(\xi \cdot h_{\mathcal{M}} + h_{\mathcal{M}}^2)$, where $C$ is a constant.*

---

[4]More rigorously, let $\mathcal{S}$ be any compact $(n-1)$-submanifold embedded in a manifold $\mathcal{M}$, the Cheeger's constant $h_{\mathcal{M}}$ is defined by the infinitum taking over all smooth $\mathcal{S}$.

That is, the increase of $\lambda$ enlarges the topological bottleneck in which the messages are often squashed. However, we identify a **negative impact** of increasing $\lambda$ from the perspective of gradient vanishing when we take a closer look. Conducting the eigenvalue decomposition, we have the following hold.

**Theorem 4.2** (Spectral Gap and Gradient Vanishing, B.1). *Suppose $t \in [t_\ell, t_{\ell+1}]$ with $1 < \ell \leq K$, the upper bound of the functional derivative $\frac{\delta u_\ell[\phi]}{\delta \phi(\boldsymbol{y})}(t, \boldsymbol{x})$ is determined by the heat kernel $H(t, \boldsymbol{x}, \boldsymbol{y})$ and a model constant $C_{model}$. Let $\psi_i$ denote the eigenfunction. The following inequality holds,*

$$\frac{\delta u_\ell[\phi]}{\delta \phi(y)}(t, \boldsymbol{x}) \leq C_{model} \cdot H(t, \boldsymbol{x}, \boldsymbol{y}) = C_{model} \cdot \sum\nolimits_{i=0}^{\infty} e^{-\lambda_i t} \psi_i(\boldsymbol{x}) \psi_i(\boldsymbol{y}). \tag{6}$$

The theorem above clarifies the relationship between the eigenvalue of the manifold heat kernel and model training, and states that the increase of $\lambda$ tends to result in gradient vanishing, undermining the effectiveness of message passing.

**On oversmoothing.** Next, we examine the impact of increasing $\lambda$ on oversmoothing via a prominent metric of Dirichlet energy [40], measuring the norm of the difference between node representations.

**Theorem 4.3** (Spectral Gap and Dirichlet Energy, B.2). *Suppose $t \in [t_\ell, t_{\ell+1}]$ with $1 < \ell \leq K$, and the initial state $u_\ell(t_\ell, \boldsymbol{x}) = \phi_\ell(\boldsymbol{x})$ has an eigenvalue decomposition such that $\phi_\ell = \sum_{i=1}^{\infty} c_{\ell i} \psi_i$, the Dirichlet energy $Dir[u_\ell(t, \cdot)] = \frac{1}{2} \int_{\mathcal{M}} |\mathrm{grad}\, u_\ell(t, \boldsymbol{x})|^2 \mathrm{vol}_{\mathcal{M}}$ decays exponentially w.r.t. to $t$ as*

$$Dir[u_\ell(t, \cdot)] \leq \frac{1}{2} C_{model} \left( \sum\nolimits_{i=1}^{\infty} c_i^2 \lambda_i e^{-2\lambda_i t} + \sum\nolimits_{i=1, \{j_k \neq i\}}^{\infty} c_{j_2}^2 \lambda_i e^{-2[\lambda_i(t-t_\ell) + \sum_{k=2}^{\ell} \lambda_{j_k}(t_k-t_{k-1})]} \right). \tag{7}$$

The theorem states that MPNN is exposed to even more severe oversmoothing by increasing $\lambda$ since the upper bound of the Dirichlet energy is increased.

**A more intuitive explanation.** Taking graph rewriting as an example, the bottleneck is widened by adding edges to allow more messages to pass through, thereby mitigating oversquashing. Meanwhile, edge deletion is required to combat oversmoothing caused by denser connections, which in turn hinders the information flow in message passing.

> **Message of the Section:** *In the global approach, an increase in the spectral gap tends to accelerate gradient vanishing and oversmoothing, thereby deteriorating the efficacy of message passing.*

## 5 Local Riemannian Geometry for the Best of Both Worlds

The theoretical insights above motivate us to seek for an alternative **local** solution. While preserving the original graph structure, we propose to adapt message passing according to the local structure or, in the language of Riemannian geometry, reshape the functional space over submanifolds. To achieve this, we establish a novel **non-homogeneous boundary condition** that is proven to simultaneously mitigate both oversquashing and oversmoothing.

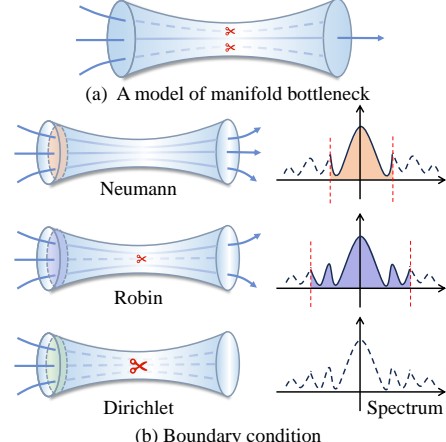

(a) A model of manifold bottleneck

Neumann

Robin

Dirichlet                    Spectrum

(b) Boundary condition

Figure 2: A visual illustration.

### 5.1 Boundary Condition over Submanifold: Local Treatment against Oversquashing

Instead of treating the manifold as a whole, we consider the heat flow over submanifold (subgraph), and introduce the notion of boundary condition to MPNN for the first time, to our best knowledge.

Specifically, a heat flow over a submanifold with boundary is described as follows

$$\begin{cases} \partial_t u(t, \boldsymbol{x}) = \Delta_g u(t, \boldsymbol{x}), & \boldsymbol{x} \in \mathrm{int}\mathcal{M}, \\ h(\boldsymbol{x}) = 0, & \boldsymbol{x} \in \partial\mathcal{M}, \end{cases} \tag{8}$$

where $\partial\mathcal{M}$ denotes the *submanifold boundary* while int$\mathcal{M}$ is the interior. The boundary is constrained on some *(homogeneous) boundary condition* $h(\boldsymbol{x}) = 0$, and there exist 3 elementary types of boundary conditions, for $\alpha \cdot \beta \geq 0$, and $\alpha, \beta$ cannot be 0 simultaneously,

$$\begin{cases} \text{Dirichlet condition} & f(\boldsymbol{x}) = 0 \\ \text{Neumann condition} & \frac{\partial f}{\partial n}(\boldsymbol{x}) = 0 \\ \text{Robin condition} & \alpha f(\boldsymbol{x}) + \beta \frac{\partial f}{\partial n}(\boldsymbol{x}) = 0 \end{cases} \tag{9}$$

where $\frac{\partial}{\partial n}$ denotes the outward normal derivative.

**Models of manifold bottlenecks.**    To formalize the effect of boundary conditions, we first introduce the models of manifold bottlenecks. Without loss of generality, we consider (1) *a simple case of cylinder*, formally described as the product manifold $\mathcal{S} = [0, L] \times \mathcal{B}_\varepsilon(r)$ of a closed interval of length $L$ and a closed $(n-1)$-ball whose radius is a constant $\varepsilon(r) = \varepsilon_0$ over $[0, L]$, and (2) *a more generic form with a functional radius* $\varepsilon(r) : [0, L] \to (0, 1)$, and an instance is sketched in Fig 2(a). Now, we demonstrate how boundary conditions affect the gradient at the bottlenecks.

**Theorem 5.1** (**Oversquashing under Boundary Conditions, B.3**). *Given the boundary of a submanifold $\mathcal{S}$ written as $\partial\mathcal{S} = \{0, L\} \times \partial\mathcal{B}_\varepsilon(s)$, we have the following claims hold:*

- *For $\varepsilon(s) = \varepsilon_0$, if the Dirichlet condition is applied, the decay of heat kernel is determined by $e^{-(\varepsilon_0^{-2}t)}$, while it is determined by $e^{-(L^{-2}t)}$ under Neumann condition.*

- *For $\varepsilon(s) = \varepsilon_0 \cosh a(s - \frac{L}{2})$, the decay of heat kernel under both the Dirichlet and Neumann conditions is related to $e^{-\left(\varepsilon_0 \cosh a \frac{L}{2}\right)^{-2}}$.*

*For both cases, the decay under Robin condition is upper bounded by the Dirichlet condition and lower bounded by the Neumann.*

**Corollary 5.2** (**Adaptivity of Robin Condition, B.4**). *For $\varepsilon(s) : [0, L] \to (0, \xi)$ and $L \gg \xi$ for a small $\xi > 0$, $\lambda^D, \lambda^N, \lambda$ are the spectral gaps of Dirichlet, Neumann and no boundary, respectively.*

- *For $\varepsilon(s) = \varepsilon_0$, we have the order $\lambda^D \geq \lambda \geq \lambda^N$ hold.*

- *For $\varepsilon(s) = \varepsilon_0 \cosh a(s - \frac{L}{2})$, if $L$ is fixed, then $\lambda^D \geq \lambda^N \geq \lambda$; if $\varepsilon_0$ is fixed, then $\lambda \geq \lambda^D \geq \lambda^N$*

Theorem 5.1 and Corollary 5.2 state that boundary conditions mitigate oversquashing by locally adjusting $\lambda$. It is achieved by controlling the amount of messages that flow out of the bottlenecks, as in Fig 2(b). Specifically, Dirichlet condition absorbs the message and only low-frequency messages can be passed. Neumann condition clears the bottleneck by reflecting all the messages. Robin condition reaches a balance between them. As $\alpha \to 0$, it degenerates to Dirichlet condition, while shifting to Neumann when $\beta \to 0$, and offers the adaptivity to adjust $\lambda$ at bottlenecks via $\alpha$ and $\beta$.

## 5.2 Nonhomogeneous Boundary Condition: Jointly Mitigating Oversmoothing

We discover that oversmoothing can be mitigated through the source term of external inputs.

**Lemma 5.3** (**Oversmoothing under Source Terms, B.5**). *For the nonhomogeneous heat equation $\partial_t u(t, \boldsymbol{x}) = \Delta_g u(t, \boldsymbol{x}) + f(t, \boldsymbol{x})$ with a source term $f(t, \boldsymbol{x}) = \sum_i B_i(t)\psi_i(\boldsymbol{x})$ and initial condition $u(0, \boldsymbol{x}) = \phi(\boldsymbol{x})$ at $t = 0$, regardless of the existence of boundary condition, there exist some parameters $\{\alpha_i > 2, \beta_i > 0\}$ satisfying the convergence condition such that*

$$\frac{dB_i}{dt} = (1 - \alpha_i + \lambda_i(\beta_i + t))(\beta_i + t)^{-\alpha_i}, \ B_i(0) = 0. \tag{10}$$

*Then, the Dirichlet energy will decay polynomially w.r.t time $t$ as*

$$Dir[u_t] = \sum_{i=1} \frac{1}{2 - \alpha_i}(\beta_i + t)^{2-\alpha_i} - \sum_{i=1} (1 + \frac{\lambda_i \beta_i}{2 - \alpha_i})\beta_i^{1-\alpha_i} \tag{11}$$

Lemma 5.3 states the source term mitigates the exponential decay of Dirichlet energy to a polynomial.

Table 1: **The boundary conditions on the graph**

| Dirichlet condition | Robin condition | Neumann condition |
|---|---|---|
| $f(u) = 0$ | $\alpha f(u) + \beta \sum_{v \in S, v \sim u}(f(u) - f(v)) = 0$ | $\sum_{v \in S, v \sim u}(f(u) - f(v)) = 0$ |

**Our solution.** We have shown the effect of boundary condition and external input on oversquashing and oversmoothing, respectively. A natural question raises that: does there exist a single operation for the best of both worlds? To this end, we establish the **nonhomogeneous boundary condition** of $h(\boldsymbol{x}) = \gamma(t, \boldsymbol{x}), \boldsymbol{x} \in \partial\mathcal{M}$, generalizing from the typical homogeneous ones of $h(\boldsymbol{x}) = 0, \boldsymbol{x} \in \partial\mathcal{M}$. While enjoying the merit of boundary condition, it can be equivalently convert into form of Lemma 5.3 to combat oversmoothing. We rigorously demonstrate the advantage of our construction below.

**Theorem 5.4** (**Equivalence, B.6**). *For the homogeneous heat equation $\partial_t u(t, \boldsymbol{x}) = \Delta_g u(t, \boldsymbol{x}), \boldsymbol{x} \in int\mathcal{M}$ with a nonhomogeneous boundary condition of $h(\boldsymbol{x}) = \gamma(t, \boldsymbol{x}), \boldsymbol{x} \in \partial\mathcal{M}$ and initial condition $u(0, \boldsymbol{x}) = \phi(\boldsymbol{x}), t = 0$, there exists a function $w(t, \boldsymbol{x}) : \mathcal{M} \to \mathbb{R}$ such that $h \circ w = \gamma$ in $\partial\mathcal{M}$. By $v = u - w$, Eq. (12) is equivalent to the nonhomogeneous heat equation satisfying Lemma 5.3:*

$$\begin{cases} \partial_t v(t, \boldsymbol{x}) = \Delta_g v(t, \boldsymbol{x}) + \Gamma(w), & \boldsymbol{x} \in int\mathcal{M}, \ t > 0 \\ h(\boldsymbol{x}) = 0, & \boldsymbol{x} \in \partial\mathcal{M}, \ t > 0, \end{cases} \tag{12}$$

*where $\Gamma(w) = \Delta_g w(t, \boldsymbol{x}) - \partial_t w(t, \boldsymbol{x})$ with converted initial condition $v(0, \boldsymbol{x}) = \phi(\boldsymbol{x}) - w(0, \boldsymbol{x})$.*

Consequently, we can simultaneously mitigate oversquashing and oversmoothing, and our construction applies to the Dirichlet, Robin, or Neumann condition.

# 6 GBN: A Provable Neural Architecture

As discussed in Sec. 4 and 5, *this calls for a neural solver to the manifold heat equation with a nonhomogeneous boundary condition.* We address this challenge with a simple yet effective Graph Boundary conditioned message passing Neural network (GBN), grounded in the Robin condition. GBN unifies the updating rules for boundary and interior nodes, and we emphasize that it is a scalable model with favorable theoretical guarantees. We begin by introducing the concept of boundary nodes.

**Definition 6.1** (Boundary Nodes). *In a graph $G$, for a subset $S$ of the vertex set $V$, the induced subgraph on $S$ has the edge set consisting of all edges of $G$ with both endpoints in $S$. The vertex boundary is $\partial S = \{u | u \sim v, v \in S\}$, where $u \sim v$ denotes $u$ is connected to $v$ in $G$.*

**MPNN under Robin Condition** Analogous to the continuous realm, the homogeneous boundary conditions are given in Table 1. Now, we construct the nonhomogeneous Robin condition on graphs,

$$\alpha_i \boldsymbol{x}_i + \beta_i \sum_{j \sim i, j \in \mathcal{V} \backslash \partial S}(\boldsymbol{x}_i - \boldsymbol{x}_j) = \gamma_i, \ i \in \partial S, \tag{13}$$

where the coefficients are given by functions with respect to node current states, i.e., $\alpha_i = \alpha(\boldsymbol{x}_i), \beta_i = \beta(\boldsymbol{x}_i), \gamma_i = \gamma(\boldsymbol{x}_i)$ for each $i \in \partial S$. Subject to Eq. (13) on the boundary nodes, we consider the nonhomogeneous heat equation for the interior nodes, whose nontrivial steady state $\mathbf{F}_S$ exists.

Accordingly, we are to solve the following equation system in the matrix form,

$$\begin{cases} \text{Interior nodes:} & \mathbf{L}_{S,S}\mathbf{X}_S + \mathbf{L}_{S,\partial S}\mathbf{X}_{\partial S} = \mathbf{F}_S, \\ \text{Boundary nodes:} & \text{diag}(\alpha_i)\mathbf{X}_{\partial S} + \text{diag}(\beta_i)\mathbf{L}_{\partial S,S}\mathbf{X}_S = \mathbf{\Gamma}_{\partial S}, \end{cases} \mathbf{L} = \begin{bmatrix} \mathbf{L}_{S,S} & \mathbf{L}_{S,\partial S} \\ \mathbf{L}_{\partial S,S} & \mathbf{L}_{\partial S,\partial S} \end{bmatrix}. \tag{14}$$

**Lemma 6.2** (**Solution to Heat Equation System, B.7**). *Let $\mathbf{D}$, $\mathbf{U}$ and $\mathbf{V}$ denote the diagonal, upper and lower matrix of $\mathbf{L}$, respectively. Accordingly to the Jacobi method, the solution is derived as,*

$$\text{diag}(\alpha_i) = \alpha(\mathbf{X}^{(k)}; \theta_\alpha), \text{diag}(\beta_i) = \beta(\mathbf{X}^{(k)}; \theta_\beta), \mathbf{\Gamma} = \gamma(\mathbf{X}^{(k)}; \theta_\gamma), \tag{15}$$

$$\mathbf{X}^{(k+1)} = \mathbf{D}^{-1}(\mathbf{V} + \mathbf{U})\mathbf{X}^{(k)} + \mathbf{D}^{-1}\mathbf{\Gamma}. \tag{16}$$

*Let $I_i := \mathbb{I}(i \in S)$ be an indicator, $\hat{d}_i = d_i(1 - I_i) + (2I_i - 1)\sum_{j \sim i} I_j$, and $p_i = \frac{\beta_i}{\alpha_i}$.*

$$[(\mathbf{D}^{-1}\mathbf{U})\mathbf{X}^{(k)}]_i = \frac{I_i}{\sqrt{\hat{d}_i}} \sum_{j \sim i} \frac{1}{\sqrt{\hat{d}_j}} x_j^{(k)}, \quad [(\mathbf{D}^{-1}\mathbf{V})\mathbf{X}^{(k)}]_i = \frac{p_i + (1 - p_i)I_i}{\sqrt{\hat{d}_i}} \sum_{j \sim i} \frac{I_j}{\sqrt{\hat{d}_j}} x_j^{(k)}. \tag{17}$$

Table 2: Comparison of layer-wise complexity. $\mathcal{V}$ and $\mathcal{E}$ denotes node and edge set, respectively.

| Models | DR | BORF | UniFilter | ProxyGap | Graph Transformer | GBN (Ours) |
|---|---|---|---|---|---|---|
| **Complexity** | $\mathcal{O}(\|\mathcal{V}\|log(\|\mathcal{V}\|))$ | $\mathcal{O}(\|\mathcal{E}\|d_{max}^3)$ | $\mathcal{O}(\|\mathcal{V}\|+\|\mathcal{E}\|)$ | $\mathcal{O}(\|\mathcal{E}\|*\|\mathcal{V}\|)$ | $\mathcal{O}(\|\mathcal{V}\|^2)$ | $\mathcal{O}(\|\mathcal{V}\|+\|\mathcal{E}\|)$ |

The updating rule is given by adding nonlinearity $\sigma$ and a learnable MLP $\varphi^{(k+1)}$,

$$\mathbf{X}^{(k+1)} = \sigma\left(\mathbf{D}^{-1}(\mathbf{V}+\mathbf{U})\varphi^{(k+1)}(\mathbf{X}^{(k)})\right) + \mathbf{D}^{-1}\mathbf{\Gamma} \tag{18}$$

Note that GBN unifies the updating rule of boundary and interior nodes via an indicator, which is jointly optimized together with other parameters.

**Computational Complexity** A pseudocode description of training GBN is given in Appendix C, whose layer-wise complexity is yielded as $\mathcal{O}(\|\mathcal{V}\|+\|\mathcal{E}\|)$. We compare with some popular models in Table 2, where $d_{max}$ is the maximum node degree.

**Local Bottleneck Adjustment** Building on a nonhomogeneous Robin condition, we locally consider each bottleneck and perform adaptive message passing. Specifically, the amount of messages flowing out of the bottleneck is regulated by the corresponding interior nodes, thereby mitigating oversquashing, while the cutoff frequency of message passing is learned in an end-to-end fashion.

**Theoretical Guarantees** Oversmoothing is mitigated through the external inputs since the Dirichlet energy does not converge to zero owing to the existence of $\gamma$. On oversquashing, we examine whether or not the Jocabian $\|\partial\boldsymbol{x}_i^{(K)}/\partial\boldsymbol{x}_j\|$ exhibits the exponential decay regarding the hop distance $K$.

**Theorem 6.3** (**Distance Independence, B.8**). *Given two nodes $v_i$ and $v_j$ with the hop distance $K$, and the updating rule in Eq. (18), we have the Jocabian satisfy*

$$\left\|\frac{\partial\boldsymbol{x}_i^{(K)}}{\partial\boldsymbol{x}_j}\right\| \leq C_K\left(\sum_{k=0}^{K}\left[\mathbf{D}^{-1}\left(\mathbf{V}+\mathbf{U}\right)\right]^k\right)_{ij}$$

*where $C_K$ is a constant related to the parameters. As $K \to \infty$, $\|\partial\boldsymbol{x}_i^{(K)}/\partial\boldsymbol{x}_j\|$ is independent of $K$.*

The theorem states that, in the limit, the Jocabian converges to a constant independent on the distance, and thus implies that oversquashing does not occur.

**Connection to Previous Models** The recent virtual node methods [45; 50] can be regarded as homogeneous Neumann condition with a single additional boundary, and have no guarantee against oversmoothing. Note that, the proposed GBN recovers the vanilla GCN when $\beta$ and $\gamma$ are eliminated.

## 7 Experiment

We compare 17 strong baselines on 7 benchmark datasets to answer the following research questions: 1) How does GBN perform on homophilic and heterophilic graphs? 2) How does each component contribute to GBN? 3) Does GBN allow for the network to go deep? 4) Does GBN support capturing long-range dependencies? **Codes** are available `https://github.com/ZhenhHuang/GBN`.

### 7.1 Experimental Setups

**Datasets & Baselines.** Experiments are conducted on a variety of datasets, including 3 homophilic graphs of WikiCS, Computer and CS [48; 19], and 4 heterophilic graphs of Texas, Wisconsin, RomanEmpire and Ratings [43]. The baselines are categorized into three groups: (1) **MPNNs**: GCN [33], GAT [66], GIN [73], hyperbolic HGCN [10] and a message diffusion GREAD [15]; (2) **Methods to improve MPNN**: G$^2$-GCN [46] and GraphNormv2 [47] against oversmoothing, SWAN [24] and MPNN+VN [50] on oversquashing, and BORF [39], DR [4], ProxyDelete [28] and UniFilter [27] to tackle both issues; (3) **Graph transformers**: sp-exphormer [49], VCR-Graphormer [21], CoBFormer [70], and GraphMamba [67]. Datasets and baselines are detailed in Appendix D.

Table 3: Node Classification in terms of ACC (%) on both homophilic and heterophilic graphs. The best results are **boldfaced**, and the runner-ups are underlined.

| Model | WikiCS | Computer | CS | Texas | Wisconsin | RomanEmpire | Ratings |
|---|---|---|---|---|---|---|---|
| GCN | $77.43_{\pm0.83}$ | $88.20_{\pm1.44}$ | $91.49_{\pm0.42}$ | $55.83_{\pm5.99}$ | $50.16_{\pm5.91}$ | $70.30_{\pm0.73}$ | $48.12_{\pm0.54}$ |
| GAT | $75.81_{\pm1.03}$ | $88.36_{\pm1.34}$ | $92.17_{\pm0.46}$ | $65.22_{\pm9.07}$ | $50.78_{\pm6.41}$ | $80.89_{\pm0.70}$ | $48.78_{\pm0.69}$ |
| GIN | $82.37_{\pm2.45}$ | $90.69_{\pm1.26}$ | $93.69_{\pm0.89}$ | $47.37_{\pm2.12}$ | $69.23_{\pm1.15}$ | $82.20_{\pm0.49}$ | $48.68_{\pm0.72}$ |
| HGCN | $74.32_{\pm0.67}$ | $89.98_{\pm1.56}$ | $90.79_{\pm0.21}$ | $62.78_{\pm1.78}$ | $55.05_{\pm1.84}$ | $80.42_{\pm0.35}$ | $47.61_{\pm2.09}$ |
| GREAD | $79.25_{\pm1.16}$ | $88.21_{\pm1.86}$ | $91.20_{\pm0.75}$ | $84.37_{\pm4.63}$ | $85.23_{\pm3.21}$ | $73.25_{\pm2.26}$ | $47.77_{\pm3.01}$ |
| DR | $77.62_{\pm0.16}$ | $89.89_{\pm0.52}$ | $90.77_{\pm0.14}$ | $70.68_{\pm1.60}$ | $70.98_{\pm1.50}$ | $71.99_{\pm0.14}$ | $48.23_{\pm0.22}$ |
| UniFilter | $84.00_{\pm0.26}$ | $86.16_{\pm2.80}$ | $95.25_{\pm0.11}$ | $78.26_{\pm1.30}$ | $69.52_{\pm1.59}$ | $74.43_{\pm0.25}$ | $49.44_{\pm0.20}$ |
| ProxyGap | $81.34_{\pm0.76}$ | $89.21_{\pm0.76}$ | $93.75_{\pm0.43}$ | $74.21_{\pm1.25}$ | $67.75_{\pm1.64}$ | $77.45_{\pm0.68}$ | $49.75_{\pm0.46}$ |
| MPNN+VN | $83.67_{\pm0.66}$ | $90.80_{\pm0.88}$ | $92.51_{\pm0.40}$ | $75.65_{\pm6.77}$ | $80.63_{\pm2.84}$ | $81.07_{\pm0.26}$ | $46.46_{\pm0.33}$ |
| GraphNormv2 | $76.93_{\pm0.45}$ | $87.26_{\pm0.16}$ | $93.80_{\pm0.15}$ | $72.07_{\pm1.56}$ | $83.01_{\pm1.13}$ | $71.79_{\pm0.34}$ | $45.80_{\pm0.06}$ |
| BORF | $85.89_{\pm0.28}$ | **$91.76_{\pm0.63}$** | $93.48_{\pm0.01}$ | $74.88_{\pm0.12}$ | $74.11_{\pm0.20}$ | $81.83_{\pm0.00}$ | $53.21_{\pm0.01}$ |
| G$^2$-GCN | $77.85_{\pm1.34}$ | $87.08_{\pm1.34}$ | $92.32_{\pm0.26}$ | $84.86_{\pm5.43}$ | $86.12_{\pm4.51}$ | $78.56_{\pm1.67}$ | $49.38_{\pm2.24}$ |
| SWAN | $79.20_{\pm1.21}$ | $88.57_{\pm0.84}$ | $90.57_{\pm0.84}$ | $76.96_{\pm7.29}$ | $73.98_{\pm2.49}$ | $73.48_{\pm0.83}$ | $49.73_{\pm3.41}$ |
| CoBFormer | $83.95_{\pm0.28}$ | $89.96_{\pm0.54}$ | $94.66_{\pm0.14}$ | $76.52_{\pm2.38}$ | $85.40_{\pm2.84}$ | $76.46_{\pm0.23}$ | $48.59_{\pm0.25}$ |
| VCR-Graphormer | $84.32_{\pm0.01}$ | $90.53_{\pm0.01}$ | $95.17_{\pm0.00}$ | $62.63_{\pm0.50}$ | $73.33_{\pm0.09}$ | $74.55_{\pm0.00}$ | $53.31_{\pm0.01}$ |
| Spexphormer | $84.53_{\pm0.15}$ | $89.73_{\pm0.39}$ | $95.19_{\pm0.15}$ | $76.96_{\pm3.65}$ | $80.95_{\pm2.84}$ | $87.54_{\pm0.14}$ | $48.61_{\pm0.32}$ |
| Graph-Mamba | $84.07_{\pm0.76}$ | $88.54_{\pm0.27}$ | $94.46_{\pm0.16}$ | $73.33_{\pm1.27}$ | $77.42_{\pm1.35}$ | $88.36_{\pm0.06}$ | $49.58_{\pm1.22}$ |
| Ours | **$86.21_{\pm0.39}$** | $91.33_{\pm0.32}$ | **$95.78_{\pm0.21}$** | **$85.01_{\pm6.51}$** | **$86.78_{\pm3.84}$** | **$89.83_{\pm0.46}$** | **$53.51_{\pm0.88}$** |

**Evaluation Metrics.** We evaluate the model performance with node classification and employ the popular metric of ACC. Each case undergoes 10 independent runs, and we report the mean with standard deviations.

Table 4: Ablation Study

| Model | CS | WikiCS | Texas | Ratings |
|---|---|---|---|---|
| GBN | **$95.78_{\pm0.21}$** | **$86.21_{\pm0.39}$** | **$85.01_{\pm6.51}$** | **$53.51_{\pm0.88}$** |
| $\gamma_0, \beta_0$ | $94.14_{\pm0.26}$ | $85.36_{\pm0.45}$ | $83.17_{\pm4.32}$ | $53.05_{\pm0.78}$ |
| $\gamma_i = 0$ | $92.26_{\pm0.58}$ | $80.01_{\pm0.34}$ | $76.27_{\pm5.34}$ | $48.23_{\pm1.03}$ |
| $\beta_i = 0$ | $93.77_{\pm0.36}$ | $81.71_{\pm0.52}$ | $82.78_{\pm5.36}$ | $50.69_{\pm0.89}$ |
| $\gamma_i, \beta_i = 0$ | $91.71_{\pm0.34}$ | $78.29_{\pm0.66}$ | $62.16_{\pm4.36}$ | $48.69_{\pm0.36}$ |
| GCN | $91.49_{\pm0.42}$ | $77.43_{\pm0.83}$ | $55.83_{\pm5.99}$ | $48.12_{\pm0.54}$ |

**Reproducibility & Model Configuration.**
The hardware is NVIDIA GeForce RTX 4090 GPU 24GB memory, and Intel Xeon Platinum 8352V CPU with 120GB RAM. In GBN, the coefficients of $\alpha$, $\beta$, and $\gamma$ are implemented as 2-layer MLP, and the model is optimized by Adam. Further details on hyperparameter settings are in Appendix E.

### 7.2 Results and Discussion

**Model Performance.** We conduct node classification on all the 7 graphs, and the results of ACC are summarized in Table 3. The hyperparameter settings of the baselines is the same as the original papers. As in Table 3, mitigating oversmoothing or oversquashing generally improves the performance of MPNNs. GBN consistently achieves the best results except for one case on the Computer dataset.

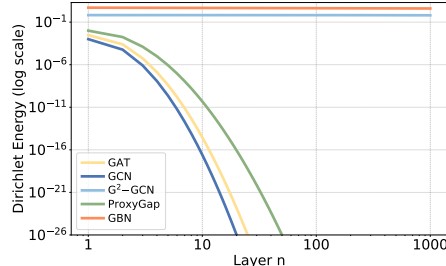

Figure 3: Dirichlet energy curve (lg-lg).

**Ablation Study.** To examine the effectiveness of each module in GBN, we design several variant models and collect the model performance in Table 4. Concretely, we replace the learnable boundary condition coefficients with fixed constants, termed as $\gamma_0, \beta_0$. The variant $\gamma_i = 0$ disables the external inputs, while $\beta_i = 0$ eliminates boundary interactions. Table 4 reports a performance decline in these variants compared to GBN, underscoring the critical role of each module. Also, under $\gamma_i, \beta_i = 0$, GBN degenerates as GCN in general.

**Deeper Network & Oversmoothing.** On oversmoothing, we evaluate the model performance and Dirichlet energy when MPNN goes deeper. Fig. 3 shows the decay of Dirichlet energy as the number of layers increases

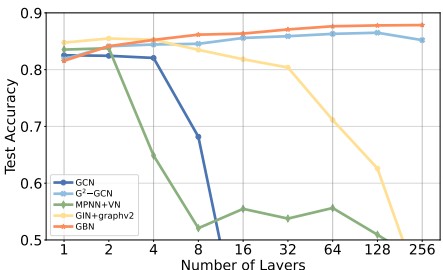

Figure 4: Node classification accuracy with respect to the number of network layers on Cora dataset.

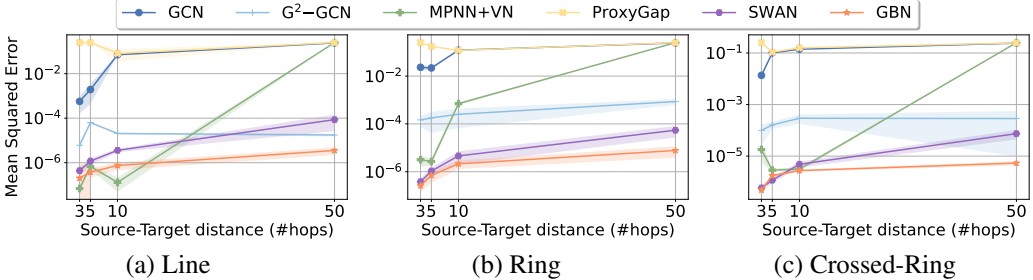

(a) Line           (b) Ring           (c) Crossed-Ring

Figure 5: Results of graph transfer task in terms of Mean Squared Error (%).

from 1 to 256 using the benchmark synthetic dataset [46; 29], detailed in Appendix E, for a fair comparison. In parallel, we present node classification performance in Fig. 4. Traditional GCN and GAT exhibit exponential decay of Dirichlet energy and a corresponding performance decline, indicating severe oversmoothing. The graph rewriting method of BORF mitigates the decay, but ultimately, node representations become indistinguishable. Both $G^2$-GCN and our proposed GBN effectively combat oversmoothing in terms of Dirichlet energy; however, $G^2$-GCN struggles to capture long-range dependencies as shown in the next paragraph. Notably, as shown in Fig. 4, *the proposed GBN does not exhibit performance degradation even the network depth exceeds* 256 *layers.*

**Long-range Dependencies & Oversquashing**
To evaluate the ability against oversquashing, we investigate the information transfer between distant nodes with the graph transfer experiment, following [24; 23]. Concretely, the experiment is conducted on 3 synthetic topologies: `line`, `ring` and `cross-ring`, where a source node labeled "1" and a target node labeled "−1" are separated by $k$ hops of randomly valued nodes. Further details are in Appendix E. The task is to switch the labels between source and target nodes through message passing. We vary the hop distance $k$ in $\{3, 5, 10, 50\}$ and present the results in Fig. 5, where the number of network layers is equal to the hop distance. As the distance increases, the messages from the expand-

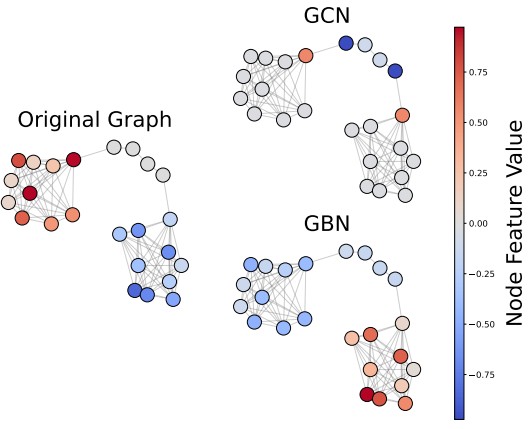

Figure 6: Case study

ing receptive field are squashed together. GCN, $G^2$-GCN and ProxyGap fail to effectively propagate the labels, revealing a limitation in capturing long-range dependencies. In contrast, the proposed GBN consistently achieves the best performance in Fig. 5.

**Case Study & Visualization** In the case study, we visualize a result of graph transfer task in Fig. 6 to further investigate the model performance. Specifically, the experiment is conducted on the graph where two complete graphs (i.e., the source on the left and the target on the right) are connected by a line (i.e., topological bottleneck). The nodes are randomly assigned in $[0, 1]$ in the source, while $[−1, 0]$ in the target. The node value in the connecting line is "0". The task is to swap the values in the source and target components. As shown in Fig. 6, GCN results in the messages being severely "squashed" in the bottleneck. In contrast, the proposed GBN allows messages to pass through the bottleneck, thanks to local bottleneck adjustment.

## 8 Conclusion

Our work presents a fresh local approach to jointly mitigate oversquashing and oversmoothing in MPNNs. Grounded in Riemannian geometry, we establish the *nonhomogeneous boundary condition* that preserves the original graph structure while adaptively adjusting the message passing paradigm to achieve the best of both worlds. Building on the Robin condition, we design a simple yet effective GBN with local bottleneck adjustment, supported by favorable theoretical guarantees. Extensive experiments demonstrate the superior performance of GBN on benchmark graphs. We further demonstrate GBN can enable deeper MPNNs by exploiting the nonhomogeneous boundary condition.

# Acknowledgement

This work is supported in part by NSFC under grants 62202164. Philip S. Yu is supported in part by NSF under grants III-2106758, and POSE-2346158.

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

# A  Notations

Table 5: Notations.

| Notation | Description |
|:---:|:---:|
| $\mathcal{M}$ | A Riemannian manifold |
| $g$ | The Riemannian metric |
| $\boldsymbol{x}$ | A point on Riemannian manifold or a vector |
| $\mathrm{vol}_{\mathcal{M}}$ | The volume form on $\mathcal{M}$ |
| $\int_{\mathcal{M}} \cdot \mathrm{vol}_{\mathcal{M}}(\boldsymbol{y})$ | The Riemannian integral w.r.t. variable $\boldsymbol{y}$ |
| $\Delta_g$ | The Laplace-Beltrami operator on $\mathcal{M}$ w.r.t. $g$ |
| grad | The Riemannian gradient |
| div | The Riemannian divergence |
| $\lambda_i, \psi_i$ | The $i$-th eigenvalue and eigenfunction of the Laplace-Beltrami operator |
| $\partial_t$ | The parital derivative w.r.t. $t$ |
| $u(t, \boldsymbol{x})$ | The solution of the heat equation |
| $H(t, \boldsymbol{x}, \boldsymbol{y})$ | The heat kernel |
| $\lambda$ | The spectral gap of a closed manifold |
| $\lambda^D$ | The spectral gap of a compact manifold with Dirichlet boundary |
| $\lambda^N$ | The spectral gap of a compact manifold with Neumann boundary |
| $\mathcal{G} = (\mathcal{V}, \mathcal{E})$ | A graph with node set $\mathcal{V}$ and edge set $\mathcal{E}$ |
| $\mathbf{X}$ | The feature matrix |
| $\mathbf{L}$ | The normalized graph Laplacian |
| $\mathbf{A}$ | The graph adjacency matrix |
| $\varphi$ | A MLP |
| $\phi$ | The initial value or function of the heat equation |
| $u, v, v_i, v_j$ | A node in graph |
| $\ell, i, j, k, m$ | The lower index |
| $\mathbf{I}$ | The identity matrix |
| $\mathcal{S}$ | The $(n-1)$-dimensional submanifold of $\mathcal{M}$ |
| Area | The volume of $\mathcal{M}$ restrict to the $(n-1)$-dimensional submanifold $\mathcal{S}$ |
| vol | The volume of $\mathcal{M}$ |
| $h_{\mathcal{M}}$ | The Cheeger's constant of $\mathcal{M}$ |
| $\xi$ | A small value variable |
| $\frac{\delta f[\phi]}{\delta \phi}$ | The variational derivative of $f$ w.r.t. $\phi$ |
| $\mathrm{Dir}[f]$ | The Dirichlet energy of function $f$ |
| $\frac{\partial}{\partial n}$ | The outward normal derivative |
| $\alpha, \beta$ | The coefficients of the Robin condition |
| $\varepsilon(\cdot)$ | The radius function |
| $\mathcal{B}_\varepsilon(r)$ | The closed ball with radius $\varepsilon(r)$ |
| $S$ | The interior node set of graph $G$ |
| $\partial S$ | The boundary node set of graph $G$ |
| $\mathbf{D}$ | The diagonal matrix for the Jacobi method |
| $\mathbf{V}$ | The lower diagonal matrix for the Jacobi method |
| $\mathbf{U}$ | The upper diagonal matrix for the Jacobi method |
| $\boldsymbol{\Gamma}_{\partial S}$ | The source term of nonhomogeneous Robin condition for boundary nodes |
| $\mathbf{F}_S$ | The steady state of heat equation for interior nodes |
| $\boldsymbol{\Gamma}$ | The concatenate for $\mathbf{F}_S$ and $\boldsymbol{\Gamma}_{\partial S}$ |

# B  Theorems, Proofs and Derivations

## B.1  Proofs of Theorem 4.2

**Theorem 4.2 (Spectral Gap and Gradient Vanishing).** *Suppose $t \in [t_\ell, t_{\ell+1}]$ with $1 < \ell <= K$, the upper bound of the functional derivative $\frac{\delta u_\ell[\phi]}{\delta \phi(\boldsymbol{y})}(t, \boldsymbol{x})$ is determined by the heat kernel $H(t, \boldsymbol{x}, \boldsymbol{y})$ and the model constant $C_{model}$,*

$$\frac{\delta u_\ell[\phi]}{\delta \phi(y)}(t, \boldsymbol{x}) \le C_{model} \cdot H(t, \boldsymbol{x}, \boldsymbol{y}) = C_{model} \cdot \sum_{i=0} e^{-\lambda_i t} \psi_i(\boldsymbol{x}) \psi_i(\boldsymbol{y}).$$

*Proof.* The algebraic operations are omitted for clarity[5], and the key steps of the derivation are shown as follows,

$$
\begin{aligned}
\frac{\delta u_\ell[\phi]}{\delta \phi(y)}(t, x) &= \int_{\mathcal{M}} \frac{\delta u_\ell[\phi_\ell]}{\delta \phi_\ell(z_\ell)}(t, x) \frac{\delta \phi_\ell[\phi]}{\delta \phi(y)} \operatorname{vol}_{\mathcal{M}}(z_\ell) \\
&= \int_{\mathcal{M}} \frac{\delta u_\ell[\phi_\ell]}{\delta \phi_\ell(z_\ell)}(t, x) \frac{\delta G_{\theta_\ell}[u_{\ell-1}]}{\delta u_{\ell-1}(t_\ell, z_\ell)} \frac{\delta u_{\ell-1}[\phi]}{\delta \phi(y)}(t_\ell, z_\ell) \operatorname{vol}_{\mathcal{M}}(z_\ell) \\
&= \int_{\mathcal{M}^\ell} \frac{\delta u_\ell[\phi_\ell]}{\delta \phi_\ell(z_\ell)}(t, x) \frac{\delta G_{\theta_\ell}[u_{\ell-1}]}{\delta u_{\ell-1}(t_\ell, z_\ell)} \left[ \prod_{i=2}^{\ell-1} \frac{\delta u_i[\varphi_i]}{\delta \varphi_i(z_i)}(t_{i+1}, z_{i+1}) \frac{\delta G_{\theta_i}[u_i]}{\delta u_{i-1}(t_i, z_i)} \right] \\
&\quad \frac{\delta u_1[\phi_1]}{\delta \phi_1(z_1)}(t_2, z_2) \frac{\delta \phi_1[\phi]}{\delta \phi(y)}(z_2) \operatorname{vol}_{\mathcal{M}^l}(\times_{i=1}^{\ell} z_i) \\
&= \int_{\mathcal{M}^l} H(t - t_\ell, x, z_\ell) \left[ \prod_{i=2}^{\ell-1} H(t_{i+1} - t_i, z_{i+1}, z_i) \right] H(t_2 - t_1, z_2, y) \\
&\quad \left[ \prod_{i=2}^{\ell} \frac{\delta G_{\theta_i}[u_{i-1}]}{\delta u_{i-1}(t_i, z_i)} \right] \operatorname{vol}_{\mathcal{M}^\ell}(\times_{i=2}^{\ell} z_i) \\
&\le H(t, x, y) \int_{\mathcal{M}^\ell} \left[ \prod_{i=2}^{\ell} \frac{\delta G_{\theta_i}[u_{i-1}]}{\delta u_{i-1}(t_i, z_i)} \right] \operatorname{vol}_{\mathcal{M}^\ell}(\times_{i=2}^{\ell} z_i) \\
&\le C_{\text{model}} H(t, x, y)
\end{aligned}
\tag{19}
$$

Since $\mathcal{M}$ is compact, by the eigen-decomposition w.r.t $H(t, x, y)$, we have

$$\frac{\delta u_l[\phi]}{\delta \phi(y)}(t, x) \le C_{\text{model}} H(t, x, y) = C_{\text{model}} \cdot \sum_{i=0}^{\infty} e^{-\lambda_i t} \psi_i(x) \psi_i(y) \tag{20}$$

$\square$

## B.2  Proofs of Theorem 4.3

**Theorem 4.3 (Spectral Gap and Dirichlet Energy).** *Suppose $t \in [t_\ell, t_{\ell+1}]$ with $1 < \ell \le K$, and the initial state $u_\ell(t_\ell, \boldsymbol{x}) = \phi_\ell(\boldsymbol{x})$ has an eigenvalue decomposition such that $\phi_\ell = \sum_{i=1}^{\infty} c_{\ell,i} \psi_i$, the Dirichlet energy $Dir[u_\ell(t, \cdot)] = \frac{1}{2} \int_{\mathcal{M}} |\operatorname{grad} u_\ell(t, \boldsymbol{x})|^2 \operatorname{vol}_{\mathcal{M}}$ decays exponentially w.r.t. to $t$ as*

$$Dir[u_\ell(t, \cdot)] \le \frac{1}{2} w_{model} \left( \sum_{i=1}^{\infty} c_i^2 \lambda_i e^{-2\lambda_i t} + \sum_{i=1, \{j_k \neq i\}}^{\infty} c_{j_2}^2 \lambda_i e^{-2[\lambda_i(t - t_\ell) + \sum_{k=2}^{\ell} \lambda_{j_k}(t_k - t_{k-1})]} \right).$$

*where $\psi_i$ denotes eigenfunctions and the $c_{\ell,i}$ are the corresponding coefficients.*

---

[5]Throughout the proofs in Part B, we demonstrate the derivation with the scalar $x$, and the same structure holds for the vector $\boldsymbol{x}$.

*Proof.* Suppose $\phi_\ell(x) \sum_{i=1} c_{\ell,i}\psi_i(x)$, where $c_{\ell,i} = \langle \phi_\ell, \psi_i \rangle$, we have

$$
\begin{aligned}
u_\ell(t,x) &= \int_{\mathcal{M}} H(t-t_\ell, x, y)\phi_\ell(y)\,\mathrm{vol}_{\mathcal{M}}(y) \\
&= \int_{\mathcal{M}} \sum_{i,j} e^{-\lambda_i(t-t_\ell)}\psi_i(x)\psi_i(y)c_{\ell,j}\phi_j(y)\,\mathrm{vol}_{\mathcal{M}}(y) \\
&= \sum_{i,j} e^{-\lambda_i(t-t_\ell)}\psi_i(x)c_{\ell,j}\int_{\mathcal{M}} \psi_i(y)\psi_j(y)\,\mathrm{vol}_{\mathcal{M}}(y) \\
&= \sum_{i=1}^{\infty} e^{-\lambda_i(t-t_\ell)}c_{\ell,i}\psi_i(x)
\end{aligned}
\tag{21}
$$

The Riemannian gradient of $u_\ell(t,\cdot)$ is

$$
\mathrm{grad}_x\, u_\ell(t,x) = \sum_{i=1}^{\infty} e^{-\lambda_i(t-t_\ell)}c_{\ell,i}\,\mathrm{grad}_x\,\psi_i
\tag{22}
$$

Then, the Dirichlet energy of $u_\ell(t,\cdot)$ will be

$$
\begin{aligned}
\mathrm{Dir}\left[u_l(t)\right] &= \frac{1}{2}\int_{\mathcal{M}} |\nabla u_\ell(t,x)|^2\,\mathrm{vol}_{\mathcal{M}} \\
&= \frac{1}{2}\int_{\mathcal{M}} \langle \nabla u_\ell, \nabla u_\ell\rangle_g\,\mathrm{vol}_{\mathcal{M}} \\
&= \frac{1}{2}\sum_{i,j} e^{-(\lambda_i+\lambda_j)(t-t_\ell)}c_{\ell,i}c_{\ell,j}\int_{\mathcal{M}} \langle \nabla_x\psi_i, \nabla_x\psi_j\rangle\,\mathrm{vol}_{\mathcal{M}} \\
&= -\frac{1}{2}\sum_{i,j} e^{-(\lambda_i+\lambda_j)(t-t_\ell)}c_{\ell,i}c_{\ell,j}\int_{\mathcal{M}} \psi_i\Delta_0\psi j\,\mathrm{vol}_{\mathcal{M}} \\
&= \frac{1}{2}\sum_{i,j} \lambda_j c_{\ell,i}c_{\ell,j}e^{-(\lambda_i+\lambda_j)(t-t_\ell)}\int_{\mathcal{M}} \psi_i\psi j\,\mathrm{vol}_{\mathcal{M}} \\
&= \frac{1}{2}\sum_{i} c_{\ell,i}{}^2\lambda_i e^{-2\lambda_i(t-t_\ell)}
\end{aligned}
\tag{23}
$$

since $\phi_\ell = G_{\theta_\ell}[u_{\ell-1}(t_\ell, x)]$, the projection coefficient on $\psi_i$ is

$$
\begin{aligned}
|c_{\ell_i}| &= |\int_{\mathcal{M}} G_{\theta_\ell}[u_{\ell-1}(t_\ell,x)]\psi_i\,\mathrm{vol}_{\mathcal{M}}| \\
&\leq \|G_{\theta_\ell}\|\|u_{\ell-1}(t_\ell)\|\|\psi_i\| \\
&= \|G_{\theta_\ell}\|\|u_{\ell-1}(t_\ell)\|
\end{aligned}
$$

From Eq. (21), we have

$$
\begin{aligned}
\|u_{\ell-1}(t_\ell)\|^2 &= \int_{\mathcal{M}} \sum_{i,j} c_{\ell-1,i}c_{\ell-1,j}e^{-(\lambda_i+\lambda_j)(t_\ell-t_{\ell-1})}\psi_i(x)\psi_j(x)\,\mathrm{vol}_{\mathcal{M}} \\
&= \sum_{j} c_{\ell-1,j}^2 \cdot e^{-2\lambda_j(t_\ell-t_{\ell-1})}
\end{aligned}
$$

$$
\Rightarrow c_{\ell_i}^2 \leq \|G_{\theta_i}\|^2 \sum_{j} c_{\ell-1,j}^2 e^{-2\lambda_j(t_\ell-t_{\ell-1})}
\tag{24}
$$

Substituting into Eq. (23), we have

$$
\mathrm{Dir}\left[u_\ell(t)\right] = \frac{1}{2}\sum_{i=1}^{\infty} c_{\ell,i}{}^2\lambda_i e^{-2\lambda_i(t-t_\ell)}
\tag{25}
$$

$$
\leq \frac{1}{2}\|G_{\theta_\ell}\|^2 \sum_{i,j} c_{\ell-1,j}^2\lambda_i e^{-2\lambda_i(t-t_\ell)-2\lambda_j(t_\ell-t_{\ell-1})}
\tag{26}
$$

By induction, we have

$$\text{Dir}\left[u_\ell(t)\right] \le \frac{1}{2}\left[\prod_{k=1}^{\ell}\|G_{\theta_k}\|^2\right]\sum_{i,j_2\cdots j_\ell}c_{j_2}^2\lambda_i e^{-2\left[\lambda_i(t-t_\ell)+\sum_{k=2}^{\ell}\lambda_{j_k}(t_k-t_{k-1})\right]} \tag{27}$$

$$= \frac{1}{2}w_{\text{model}}\left[\sum_i c_i^2\lambda_i e^{-2\lambda_i t} + \sum_{\substack{i,j_2\neq i\\ \cdots\\ j_\ell\neq i}}c_{j_2}^2\lambda_i e^{-2\lambda_i(t-t_\ell)+\sum_{k=2}^{\ell}\lambda_{j_k}(t_k-t_{k-1})}\right] \tag{28}$$

$$\tag{29}$$

where $c_{j_2} = \langle\phi,\psi_{j_2}\rangle$. $\qquad\qquad\qquad\qquad\qquad\qquad\qquad\qquad\qquad\qquad\qquad\qquad\square$

## B.3   Proofs of Theorem 5.1

**Theorem 5.1 (Oversquashing under Boundary Conditions).** *Given the boundary of a submanifold $\mathcal{S}$ written as $\partial\mathcal{S} = \{0, L\}\times\partial\mathcal{B}_\varepsilon(s)$, we have the following claims hold:*

- *For $\varepsilon(s) = \varepsilon_0$, if the Dirichlet condition is applied, the decay of heat kernel is determined by $e^{-(\varepsilon_0^{-2}t)}$, while it is determined by $e^{-(L^{-2}t)}$ under Neumann condition.*
- *For $\varepsilon(s) = \varepsilon_0\cosh a(s - \frac{L}{2})$, the decay of heat kernel under both the Dirichlet and Neumann conditions is related to $e^{-\left(\varepsilon_0\cosh a\frac{L}{2}\right)^{-2}}$.*

*For both cases, the decay under Robin condition is upper bounded by the Dirichlet condition and lower bounded by the Neumann condition.*

*Proof.* Consider $\mathcal{S} = \bigcup_{s\in[0,L]}\{s\}\times\mathcal{B}_\varepsilon(s)$, where $\varepsilon(r) : [0, L]\to(0,\delta)$ with $\delta\in(0,1)$. Denote the Laplace-Beltrami operator on $[0, L]$ and $\mathcal{B}_\varepsilon(s)$ are $\Delta_\mathcal{R}$ and $\Delta_{\mathcal{B}_\varepsilon,s}$. Since $\Delta_\mathcal{S} = \Delta_\mathcal{R}\otimes I_{\mathcal{B}_\varepsilon,s} + I_{[0,L]}\otimes\Delta_{\mathcal{B}_\varepsilon,s}$, for eigenfunction $\psi^\mathcal{R}$ of $\Delta_R$ and $\psi^{\mathcal{B}_\varepsilon,s}$ for $\Delta_{B_\varepsilon,s}$,

$$\Delta_\mathcal{S}(\psi_m^\mathcal{R}\otimes\psi_\ell^{\mathcal{B}_\varepsilon,s}) = \Delta_\mathcal{R}\psi_m^\mathcal{R}\otimes\psi_\ell^{\mathcal{B}_\varepsilon,s} + \psi_m^\mathcal{R}\otimes\Delta_{\mathcal{B}_\varepsilon,s}\psi_\ell^{\mathcal{B}_\varepsilon,s}$$
$$= (\mu_m^\mathcal{R} + \mu_\ell^{\mathcal{B}_\varepsilon,s})\psi_m^\mathcal{R}\otimes\psi_\ell^{\mathcal{B}_\varepsilon,s} \tag{30}$$

**The first case**   Now, we consider that $\varepsilon(s) = \varepsilon_0$, where $\varepsilon_0\in(0,\delta)$ is a constant. First, we solve the eigenvalue problem on $\mathcal{R} = [0, L]$ with boundary condition:

On the Dirichlet boundary condition, the eigenvalue problem equation is

$$\begin{cases} \frac{d^2}{dx^2}\psi^\mathcal{R} = -\mu^\mathcal{R}\psi^\mathcal{R}, & x\in(0, L)\\ \psi^\mathcal{R}(0) = \psi^\mathcal{R}(L) = 0 & x\in\{0, L\} \end{cases} \tag{31}$$

$$\Rightarrow \mu_m^\mathcal{R} = \left(\frac{m\pi}{L}\right)^2, \psi_m^\mathcal{R}(x) = \sin\left(\frac{m\pi}{L}x\right) \tag{32}$$

On the Neumann boundary condition, the eigenvalue problem equation is

$$\begin{cases} \frac{d^2}{dx^2}\psi^\mathcal{R} = -\mu^\mathcal{R}\psi^\mathcal{R}, & x\in(0, L)\\ \frac{d}{dx}\psi^\mathcal{R}|_{x=0} = -\frac{d}{dx}\psi^\mathcal{R}|_{x=L} = 0 & x\in\{0, L\} \end{cases} \tag{33}$$

$$\Rightarrow \mu_m^\mathcal{R} = \left(\frac{m\pi}{L}\right)^2, \psi_m^\mathcal{R}(x) = \cos\left(\frac{m\pi}{L}x\right) \tag{34}$$

For the eigenvalue problem in $\mathcal{B}_\varepsilon(s)$, the sectional surface is a closed ball with radius $\varepsilon_0$. The Laplace-Beltrami operator $\Delta_{\mathcal{B},\varepsilon_0} = \frac{1}{\varepsilon_0^2}\Delta_{\mathcal{B},1}$, where $\Delta_{\mathcal{B},1}$ is the Laplace-Beltrami operator on the unit ball, under the polar coordinate $(r,\theta), r\in[0,1], \theta\in\mathbb{S}^{n-1}$,

$$\Delta_{\mathcal{B},1} = \frac{1}{r^{n-1}}\frac{\partial}{\partial r}\left(r^{n-1}\frac{\partial}{\partial r}\right) + \frac{1}{r^2}\Delta_{\mathbb{S}^{n-1}} \tag{35}$$

Using separate variable method, let $\psi^{B_1} = R(r)\Theta(\theta)$, the eigenvalue problem equation is:

$$\frac{1}{r^{n-1}} \frac{d}{dr}\left(r^{n-1}\frac{dR}{dr}\right)\theta + \frac{R}{r^2}\Delta_{\mathbb{S}^{n-1}}\Theta = -\mu^{B,1}R\Theta \tag{36}$$

Dividing $\Theta$ and let $\Theta(\theta) = Y_k(\theta)$, where $Y_k(\theta)$ is the spherical harmonics such that

$$\Delta_{\mathbb{S}^{n-1}}Y_k = -k(k+n-2)Y_k \tag{37}$$

Denote $\lambda_k^\theta = k(k+n-2)$, we have

$$r^2\frac{dR^2}{dr^2} + (n-1)r\frac{dR}{dr} + (\mu^{B,1}r^2 - \lambda^\theta)R = 0 \tag{38}$$

Let $R(r) = r^{-\frac{n-2}{2}}w(r)$, then

$$r^2\frac{d^2w}{dr^2} + r\frac{dw}{dr} + (\mu^{B,1}r^2 - (k+\frac{n-2}{2})^2)w = 0 \tag{39}$$

Set $z = \sqrt{\mu^{B,\infty}}r$, we have the Bessel's different equation:

$$z^2\frac{d^2w}{dz^2} + z\frac{dw}{dz} + (z^2 - (k+\frac{n-2}{2})^2)w = 0 \tag{40}$$

Thus, $R(r) = r^{-\frac{n-2}{2}}J_\nu\sqrt{\mu^{B,1}}r$, where $\nu = k + \frac{n-2}{2}$ is the order of the Bessel function $J_\nu$.

For the Dirichlet boundary condition, it satisfies

$$R(1) = 0 \Rightarrow \mu_\ell = j_{\nu,\ell}^2, \tag{41}$$

where $j_{\nu,\ell}$ is the $\ell$-th zeros of $J_\nu$. The eigenfunction and eigenvalues of $\mathcal{B}_{\varepsilon_0}$ is:

$$\mu_\ell^{B_{\varepsilon_0}} = \left(\frac{j_{\nu,\ell}}{\varepsilon_0}\right)^2, \quad \psi_{\ell,k}^{B_{\varepsilon_0}} = r^{-\frac{n-2}{2}}J_\nu\left(\frac{j_{\nu,\ell}}{\varepsilon_0}r\right)Y_k(\theta) \tag{42}$$

For the Neumann boundary condition, it satisfies

$$R'(1) = 0 \Rightarrow \mu_\ell^{B,1} = j_{\nu,\ell}'^2, \tag{43}$$

where $j_{\nu,\ell}'$ is the $\ell$-th zeros of the derivative of $j_\nu$. The eigenfunction and eigenvalues of $\mathcal{B}_{\varepsilon_0}$ is:

$$\mu_\ell^{B_{\varepsilon_0}} = \left(\frac{j_{\nu,\ell}'}{\varepsilon_0}\right)^2, \quad \psi_{\ell,k}^{B_{\varepsilon_0}} = r^{-\frac{n-2}{2}}J_\nu\left(\frac{j_{\nu,\ell}'}{\varepsilon_0}r\right)Y_k(\theta) \tag{44}$$

Above all, the eigenvalues of $\mathcal{S} = [0,L] \times \mathcal{B}_{\varepsilon_0}$ are

$$\lambda_{m,\ell}^D = \left(\frac{m\pi}{L}\right)^2 + \left(\frac{j_{\nu,\ell}}{\varepsilon_0}\right)^2 \tag{45}$$

$$\lambda_{m,\ell}^N = \left(\frac{m\pi}{L}\right)^2 + \left(\frac{j_{\nu,\ell}'}{\varepsilon_0}\right)^2 \tag{46}$$

Since for the Dirichlet boundary condition, the spectral gap is $\lambda^D = \left(\frac{\pi}{L}\right)^2 + \left(\frac{j_{\nu,1}}{\varepsilon_0}\right)^2$, the decay of $e^{-\lambda^D} = e^{-\left(\frac{\pi}{L}\right)^2 - \left(\frac{j_{\nu,1}}{\varepsilon_0}\right)^2}$ is determined by $e^{-\varepsilon_0^{-2}}$, for the Neumann boundary condition, the spectral gap is $\lambda^N = \left(\frac{\pi}{L}\right)^2 + \left(\frac{j_{0,0}'}{\varepsilon_0}\right)^2 = \left(\frac{\pi}{L}\right)^2$, the decay of $e^{-\lambda^N}$ is determined by $e^{-L^{-2}}$.

**The second case** Consider the radius function $\varepsilon(s) = \varepsilon_0\frac{(e^{a(s-\frac{L}{2})}+e^{-a(s-\frac{L}{2})})}{2} = \varepsilon_0\cosh a(s-\frac{L}{2})$ for $a > 0, s \in [0,L]$. Since the radius of $\mathcal{B}$ depends on the variable in $\mathcal{R}$, we can not overlay the solution on each component to get the eigenvalues. But the separate variable is still working. Let $l = s - \frac{L}{2}$, the Laplacian-Beltrami operator on $\mathcal{S}$ is

$$\Delta_\mathcal{S} = \frac{\partial^2}{\partial l^2} + \frac{1}{\varepsilon_0\cosh^2 al}\Delta_{\mathcal{B},1} \tag{47}$$

Thus, let $\psi^{\mathcal{S}} = X(l)Y(r,\theta)$, the eigenvalue equation is:

$$\frac{d^2 X}{dl^2}Y + \frac{1}{\varepsilon_0^2 \cosh^2(al)}X\Delta_{\mathcal{B},1}Y = -\lambda^{\mathcal{S}}XY \tag{48}$$

Dividing $XY$, we have

$$\frac{1}{X}\frac{d^2 X}{dl^2} + \frac{1}{\varepsilon_0^2 \cosh^2(al)}\frac{\Delta_{\mathcal{B},1}Y}{Y} = -\lambda^{\mathcal{S}} \tag{49}$$

Set $Y$ to be the eigenfunction of $\Delta_{\mathcal{B},1}$ as in the first case, we have

$$\frac{d^2 X}{dl^2} + \left(\lambda^{\mathcal{S}} - \frac{\mu^{\mathcal{B},1}}{\varepsilon_0^2 \cosh^2(al)}\right)X = 0 \tag{50}$$

Take $z = \tanh al$,

$$\frac{d}{dl} = \frac{a}{\cosh^2(al)}\frac{d}{dz} = a(1-z^2)\frac{d}{dz}$$

$$\frac{d^2}{dl^2} = a^2(1-z^2)^2\frac{d^2}{dz^2} - 2a^2 z(1-z^2)\frac{d}{dz}$$

$$\Rightarrow a^2(1-z^2)^2\frac{d^2 x}{dz^2} - 2a^2 z(1-z^2)\frac{dx}{dz} + \left(\lambda^{\mathcal{S}} - \frac{\mu^{\mathcal{B},1}(1-z^2)}{\varepsilon_0^2}\right)X = 0$$

$$\Rightarrow (1-z^2)\frac{d^2 X}{dz^2} - 2z\frac{dX}{dz} + \left(\frac{\lambda^{\mathcal{S}}}{a^2(1-z^2)} - \frac{\mu^{\mathcal{B},1}}{a^2\varepsilon_0^2}\right)X = 0, \tag{51}$$

which is the associated Legendre differential equation:

$$(1-z^2)\frac{d^2 X}{dz^2} - 2z\frac{dX}{dz} + \left[\ell(\ell+1) - \frac{m^2}{1-z^2}\right]X = 0 \tag{52}$$

with $\ell(\ell+1) = -\frac{\mu^{\mathcal{B},1}}{a^2\varepsilon_0^2}$, $m^2 = -\frac{\lambda^{\mathcal{S}}}{a^2}$, where $\ell, m$ are the degree and order of the associated Legendre polynomial, respectively.

We obtain the solution

$$X(z) = C_1 P_\ell^m(z) + C_2 Q_\ell^m(z)$$
$$\Rightarrow X(l) = C_1 P_\ell^m(\tanh(al)) + C_2 Q_\ell^m(\tanh(al)) \tag{53}$$

where $P_\ell^m$ is the associated Legendre polynomials, $Q_\ell^m$ is Legendre function of the second kind.

Therefore, for the Dirichlet boundary condition,

$$X(0)Y(\varepsilon_0 \cosh a\frac{L}{2},\theta) = X(L)Y(\varepsilon_0 \cosh a\frac{L}{2},\theta) = 0 \tag{54}$$

We find that the spectral gap $\lambda^D$ mainly relates to $\left(\frac{j_{\nu,1}}{\varepsilon_0 \cosh a\frac{L}{2}}\right)^2$, since $\varepsilon_0 \ll L$, we have

$$\lambda^D \sim \left(\frac{j_{\nu,1}}{\varepsilon_0 \cosh a\frac{L}{2}}\right)^2 \tag{55}$$

for the Neumann boundary condition,

$$X'(0)Y(\varepsilon_0 \cosh a\frac{L}{2},\theta) + X(0)Y'(\varepsilon_0 \cosh a\frac{L}{2},\theta) = 0$$
$$X'(L)Y(\varepsilon_0 \cosh a\frac{L}{2},\theta) + X(L)Y'(\varepsilon_0 \cosh a\frac{L}{2},\theta) = 0, \tag{56}$$

the spectral gap $\lambda^N$ mainly relates to $\left(\frac{j_{\nu,0}}{\varepsilon_0 \cosh a\frac{L}{2}}\right)^2 + \left(\frac{j'_{0,0}}{\varepsilon_0 \cosh a\frac{L}{2}}\right)^2$, meaning that

$$\Rightarrow \lambda^N \sim \left(\frac{j_{\nu,0}}{\varepsilon_0 \cosh a\frac{L}{2}}\right)^2 \tag{57}$$

$\square$

## B.4 Proofs of Corollary 5.2

**Corollary 5.2 (Adaptivity of Robin Condition).** *For $\varepsilon(s) : [0, L] \to (0, \delta)$ and $L \gg \delta$ for a small $\delta > 0$, the spectral gaps $\lambda^D, \lambda^N, \lambda$ of the Dirichlet condition, Neumann condition, and no boundary have the following order:*

- *For $\varepsilon(s) = \varepsilon_0$, $\lambda^D \geq \lambda \geq \lambda^N$.*
- *For $\varepsilon(s) = \varepsilon_0 \cosh a(s - \frac{L}{2})$, if $L$ is fixed, then $\lambda^D \geq \lambda^N \geq \lambda$; if $\varepsilon_0$ is fixed, then $\lambda \geq \lambda^D \geq \lambda^N$*

*Proof.* From Cheeger and Buser's inequality (Theorem 4.1), if the manifold is closed, the spectral gap of its Laplace-Beltrami operator $\lambda \sim \inf \varepsilon^{n-1}$ when taking $\mathcal{S}$ as a submanifold embedded in a closed manifold.

For $\varepsilon(s) = \varepsilon_0$, since $\varepsilon_0$ is small, $\varepsilon_0^{-2}$ will be large, and $\varepsilon_0^{n-1}$ will be small, thus we have the order

$$\lambda^D \geq \lambda \geq \lambda^N$$

For $\varepsilon(s) = \varepsilon_0 \cosh a(s - \frac{L}{2})$, from the proofs of Theorem 5.1, since $j_{\nu,1} > j_{\nu,0}$, we have $\lambda^D > \lambda^N$. If $L$ is fixed, since $\varepsilon_0^{-2}$ will be large, $\lambda^D \geq \lambda^N \geq \lambda$; if $\varepsilon_0$ is fixed, since $L \gg \varepsilon_0$, $\cosh^{-2} a\frac{L}{2}$ will be much more smaller than $\varepsilon_0^{n-1}$, then $\lambda \geq \lambda^D \geq \lambda^N$ $\qquad\square$

## B.5 Proofs of Lemma 5.3

**Lemma 5.3 (Oversmoothing under Source Terms).** *For the nonhomogeneous heat equation $\partial_t u(t, \boldsymbol{x}) = \Delta_g u(t, \boldsymbol{x}) + f(t, \boldsymbol{x})$ with a source term $f(t, \boldsymbol{x}) = \sum_i B_i(t)\psi_i(\boldsymbol{x})$ and initial condition $u(0, \boldsymbol{x}) = \phi(\boldsymbol{x})$ at $t = 0$, regardless of the existence of boundary condition, there exist some parameters $\{\alpha_i > 2, \beta_i > 0\}$ satisfying the convergence condition such that*

$$\frac{dB_i}{dt} = (1 - \alpha_i + \lambda_i(\beta_i + t))(\beta_i + t)^{-\alpha_i}, \; B_i(0) = 0.$$

*Then, the Dirichlet energy will decay polynomially w.r.t time $t$ as*

$$Dir[u_t] = \sum_{i=1} \frac{1}{2 - \alpha_i}(\beta_i + t)^{2-\alpha_i} - \sum_{i=1}(1 + \frac{\lambda_i \beta_i}{2 - \alpha_i})\beta_i^{1-\alpha_i}.$$

*Proof.* Consider the nonhomogeneous heat equation:

$$\begin{cases} \partial_t u(t, x) = \Delta u(t, x) + f(t, x), & x \in \text{int}\mathcal{M}, t > 0 \\ h(x) = 0, & x \in \partial\mathcal{M}, t > 0 \\ u(0, x) = \phi(x), & x \in \mathcal{M}, t = 0 \end{cases} \tag{58}$$

The solution is

$$u(t, x) = \int_0^t \int_{\mathcal{M}} H(t - s, x, y)f(s, y) \text{vol}_{\mathcal{M}}(y)ds + \int_{\mathcal{M}} H(t, x, y)\phi(y) \text{vol}_{\mathcal{M}}(y) \tag{59}$$

Let $f(t, x) = \sum_i \beta_i(t)\psi_i(x)$, $\beta_i(t) = \langle f_t, \psi_i \rangle$, $\phi(x) = \sum_i c_i \psi_i(x)$, and $c_i = \langle \phi, \psi_i \rangle$. Define

$$u^1(t, x) := \int_0^t \int_{\mathcal{M}} H(t - s, x, y)f(s, y) \text{vol}_{\mathcal{M}}(y)ds \tag{60}$$

$$u^2(t, x) := \int_{\mathcal{M}} H(t, x, y)\phi(y) \text{vol}_{\mathcal{M}}(y) \tag{61}$$

Then

$$u^1(t, x) = \int_0^t \int_{\mathcal{M}} \left[\sum_{i=0}^{\infty} e^{-\lambda_i(t-s)}\psi_i(x)\psi_i(y)\right] \left[\sum_j \beta_j(s)\psi_j(y)\right] \text{vol}_{\mathcal{M}}(y)ds$$

$$= \int_0^t \sum_{i=0}^{\infty} e^{-\lambda_i(t-s)}\beta_i(s)\psi_i(x)ds$$

$$= \sum_{i=0}^{\infty} e^{-\lambda_i t} \int_0^t e^{\lambda_i s}\beta_i(s)ds\psi_i(x) \tag{62}$$

$$u^2(t, x) = \sum_{i=0}^{\infty} e^{-\lambda_i t} c_i \psi_i(x) \int_{\mathcal{M}} \psi_i(y)\psi_i(y)\, \mathrm{vol}_{\mathcal{M}}(y)$$

$$= \sum_{i=0}^{\infty} c_i e^{-\lambda_i t} \psi_i(x) \tag{63}$$

$$\tag{64}$$

For Dirichlet energy, we have

$$\mathrm{Dir}[u_t] = \frac{1}{2}\int_{\mathcal{M}} \|\mathrm{grad}\, u_t\|^2\, \mathrm{vol}_{\mathcal{M}} = \mathrm{Dir}[u_t^1] + \mathrm{Dir}[u_t^2] + \int_{\mathcal{M}} \langle \mathrm{grad}\, u_t^1, \mathrm{grad}\, u_t^2 \rangle_g\, \mathrm{vol}_{\mathcal{M}} \tag{65}$$

For each item, apply the divergence theorem or Green's first identity,

$$\mathrm{Dir}[u_t^1] = -\frac{1}{2}\int_{\mathcal{M}} u_t^1 \Delta_g u_t^1\, \mathrm{vol}_{\mathcal{M}} = \frac{1}{2}\sum_{i=1}^{\infty} \frac{1}{\lambda_i}\left[ B_i(t) - B_i(0)e^{-\lambda_i t} - e^{-\lambda_i t}\int_0^t e^{\lambda_i s}\frac{dB_i}{ds}\, ds \right] \tag{66}$$

$$\mathrm{Dir}[u_t^2] = -\frac{1}{2}\int_{\mathcal{M}} u_t^2 \Delta u_t^2\, \mathrm{vol}_{\mathcal{M}} = \sum_{i=1}^{\infty} c_i^2 \lambda_i e^{-\lambda_i t} \tag{67}$$

$$\int_{\mathcal{M}} \langle \mathrm{grad}\, u_t^1, \mathrm{grad}\, u_t^2 \rangle\, d\,\mathrm{vol}_{\mathcal{M}} = -\int_{\mathcal{M}} u_t^2 \Delta_g u_t^1\, \mathrm{vol}_{\mathcal{M}}$$

$$= \sum_{i=1}^{\infty} c_i e^{-\lambda_i t}\left[ \beta_i(t) - \beta_i(0)e^{-\lambda_i t} - e^{-\lambda_i t}\int_0^t e^{\lambda_i s}\frac{d\beta_i}{ds}\, ds \right] \tag{68}$$

If $B_i(t)$ satisfies

$$\begin{cases} B_i(0) = 0 \\ \frac{dB_i}{dt} = (1 - a_i + \lambda_i(b_i + t))(b_i + t)^{-a_i} \end{cases} \tag{69}$$

for $0 < b_i < 1, a_i > 2$, i.e.,

$$B_i(t) = \frac{\lambda_i(b_i + t)^{2-a_i}}{2 - a_i} + (b_i + t)^{1-a_i} - \left(1 + \frac{\lambda_i b_i}{2 - a_i}\right) b_i^{1-a_i} \tag{70}$$

Then,

$$E[u_t^1] = \sum_{i=1}^{\infty} \frac{1}{\lambda_i}\left[ B_i(t) - e^{-\lambda_i t}\int_0^t e^{\lambda_i s}\frac{dB_i}{ds}\, ds \right]$$

$$= \sum_{i=1}^{\infty} \frac{1}{2 - a_i}(b_i + t)^{2-a_i} - \left(1 + \frac{\lambda_i b_i}{2 - a_i}\right) b_i^{1-a_i} \tag{71}$$

So the decay of the Dirichlet energy is mainly determined by

$$\sum_{i=1}^{\infty} \frac{1}{2 - a_i}(b_i + t)^{2-a_i}$$

which is polynomially decreasing. $\qquad\square$

### B.6 Proofs of Theorem 5.4

**Theorem 5.4 (Equivalence).** *For the homogeneous heat equation $\partial_t u(t, \boldsymbol{x}) = \Delta_g u(t, \boldsymbol{x}), \boldsymbol{x} \in int\mathcal{M}$ with a nonhomogeneous boundary condition of $h(\boldsymbol{x}) = \gamma(t, \boldsymbol{x}), \boldsymbol{x} \in \partial\mathcal{M}$ and initial condition $u(0, \boldsymbol{x}) = \phi(\boldsymbol{x}), t = 0$, there exists a function $w(t, \boldsymbol{x}) : \mathcal{M} \to \mathbb{R}$ such that $h \circ w = \gamma$ in $\partial\mathcal{M}$. By $v = u - w$, Eq. (12) is equivalent to the nonhomogeneous heat equation satisfying Lemma 5.3:*

$$\begin{cases} \partial_t v(t, \boldsymbol{x}) = \Delta_g v(t, \boldsymbol{x}) + \Gamma(w), & \boldsymbol{x} \in int\mathcal{M}, \ t > 0 \\ h(\boldsymbol{x}) = 0, & \boldsymbol{x} \in \partial\mathcal{M}, \ t > 0, \\ v(0, \boldsymbol{x}) = \phi(\boldsymbol{x}) - w(0, \boldsymbol{x}), & \boldsymbol{x} \in \mathcal{M}, \ t = 0 \ \textit{(The initial condition)}, \end{cases}$$

*where $\Gamma(w) = \Delta_g w(t, \boldsymbol{x}) - \partial_t w(t, \boldsymbol{x})$.*

*Proof.* Introducing an auxiliary function $v = u - w$ such that $h \circ w = 0$ in $\partial \mathcal{M}$. Then,

$$\begin{cases} \Delta_g v = \Delta_g u - \Delta_g w \\ \partial_t v = \partial_t u - \partial_t w \end{cases}$$

$$\Rightarrow \partial_t v(t, x) = \Delta_g v(t, x) + \Delta_g w(t, x) - \partial_t w(t, x) \tag{72}$$

Given the linearity of all three types of boundary condition, we define

$$\Gamma(w) = \Delta_g w(t, x) - \partial_t w(t, x) \tag{73}$$

which satisfies the condition in Lemma 5.3 on $\text{int}\mathcal{M}$.

The equation can be converted into the form in Lemma 5.3,

$$\begin{cases} \partial_t v(t, x) = \Delta_g v(t, x) + \Gamma(w), & x \in \text{int}\mathcal{M}, \ t > 0 \\ v(0, x) = \phi(x) - w(0, x), & x \in \mathcal{M}, \ t = 0 \\ h(x) = 0, & x \in \partial\mathcal{M}, \ t > 0 \end{cases} \tag{74}$$

$\square$

## B.7 Proofs of Lemma 6.2

**Lemma 6.2 (Solution to Heat Equation System).** *Let $\mathbf{D}$, $\mathbf{U}$ and $\mathbf{V}$ denote the diagonal, upper and lower matrix of $\mathbf{L}$, respectively. Accordingly to the Jacobi method, the matrix form for the $k$-th iteration is derived as follows,*

$$\text{diag}(\alpha_i) = \alpha(\mathbf{X}^{(k)}; \theta_\alpha), \text{diag}(\beta_i) = \beta(\mathbf{X}^{(k)}; \theta_\beta), \mathbf{\Gamma} = \gamma(\mathbf{X}^{(k)}; \theta_\gamma), \tag{75}$$

$$\mathbf{X}^{(k+1)} = \mathbf{D}^{-1}(\mathbf{V} + \mathbf{U})\mathbf{X}^{(k)} + \mathbf{D}^{-1}\mathbf{\Gamma}. \tag{76}$$

*Let $I_i := \mathbb{I}(i \in S)$ be an indicator, $\hat{d}_i = d_i(1 - I_i) + (2I_i - 1)\sum_{j \sim i} I_j$, and $p_i = \frac{\beta_i}{\alpha_i}$.*

$$[(\mathbf{D}^{-1}\mathbf{U})\mathbf{X}^{(k)}]_i = \frac{I_i}{\sqrt{\hat{d}_i}} \sum_{j \sim i} \frac{1}{\sqrt{\hat{d}_j}} x_j^{(k)}, \quad [(\mathbf{D}^{-1}\mathbf{V})\mathbf{X}^{(k)}]_i = \frac{p_i + (1 - p_i)I_i}{\sqrt{\hat{d}_i}} \sum_{j \sim i} \frac{I_j}{\sqrt{\hat{d}_j}} x_j^{(k)}. \tag{77}$$

*Proof.* Merging the boundary condition and the steady solution, we have

$$\begin{bmatrix} \mathbf{L}_{S,S} & \mathbf{L}_{S,\partial S} \\ \text{diag}(\beta_i)\mathbf{L}_{\partial S,S} & \text{diag}(\alpha_i)\mathbf{I}_{\partial S,\partial S} \end{bmatrix} \begin{bmatrix} \mathbf{X}_S \\ \mathbf{X}_{\partial S} \end{bmatrix} = \begin{bmatrix} \mathbf{F}_S \\ \mathbf{\Gamma}_{\partial S} \end{bmatrix} = \mathbf{\Gamma} \tag{78}$$

Now, we use the Jacobi method to solve this nonlinear equation. The diagonal matrix is

$$\mathbf{D} = \begin{bmatrix} \mathbf{I}_{S,S} & \mathbf{0}_{S,\partial S} \\ \mathbf{0}_{\partial S,S} & \text{diag}(\alpha_i)\mathbf{I}_{\partial S,\partial S} \end{bmatrix} \tag{79}$$

The lower diagonal and upper diagonal matrices are

$$\mathbf{V} = \begin{bmatrix} \hat{\mathbf{A}}_{S,S}^{low} & \mathbf{0}_{S,\partial S} \\ -\text{diag}(\beta_i)\mathbf{L}_{\partial S,S} & \mathbf{0}_{\partial S,\partial S} \end{bmatrix}, \mathbf{U} = \begin{bmatrix} \hat{\mathbf{A}}_{S,S}^{up} & -\mathbf{L}_{S,\partial S} \\ \mathbf{0}_{\partial S,S} & \mathbf{0}_{\partial S,\partial S} \end{bmatrix} \tag{80}$$

For the $k$-th iteration,

$$\text{diag}(\alpha_i^k) = \alpha(\mathbf{X}^k; \theta_k), \text{diag}(\beta_i^k) = \beta(\mathbf{X}^k; \omega_k), \mathbf{\Gamma} = \gamma(\mathbf{X}^k; \eta_k) \tag{81}$$

$$\mathbf{X}^{k+1} = \mathbf{D}^{-1}(\mathbf{V} + \mathbf{U})\mathbf{X}^k + \mathbf{D}^{-1}\mathbf{\Gamma} \tag{82}$$

The $(i, j)$-entry of $\mathbf{D}^{-1}\mathbf{V}$ is

$$(\mathbf{D}^{-1}\mathbf{V})_{ij} = \begin{cases} \frac{1}{\sqrt{d_i^S d_j^S}} & , i \sim j, i \in S, j \in S \\ 0 & , i \sim j, i \in \partial S, j \in \partial S \\ \frac{\beta_i}{\alpha_i \sqrt{d_i^{\partial S} d_j^S}} & , i \sim j, i \in \partial S, j \in S \\ 0 & , else \end{cases} \tag{83}$$

The $(i, j)$-entry of $\mathbf{D}^{-1}\mathbf{U}$ is

$$(\mathbf{D}^{-1}\mathbf{U})_{ij} = \begin{cases} \frac{1}{\sqrt{d_i^S d_j^S}} & , i \sim j, i \in S, j \in S \\ 0 & , i \sim j, i \in \partial S, j \in \partial S \\ \frac{1}{\sqrt{d_i^S d_j^{\partial S}}} & , i \sim j, i \in S, j \in \partial S \\ 0 & , else \end{cases} \tag{84}$$

$\square$

## B.8 Proofs of Theorem 6.3

Before proof, we first begin with the following lemma.

**Lemma B.1** ([41]). *For the matrix function* $[\mathbf{T}(\mathbf{X})]_{ij} := T_{ij}(\boldsymbol{x}_i)$. *If the spectral radius* $\rho[\mathbf{T}(\mathbf{X})] < 1$, *then*

$$\sum_{k=0}^{\infty} \mathbf{T}(\mathbf{X})^k = (\mathbf{I} - \mathbf{T}(\mathbf{X}))^{-1} \tag{85}$$

**Theorem 6.3 (Distance Independence).** *Given two nodes* $v_i$ *and* $v_j$ *with distance* $K$, *the measure of information transport*

$$\left\| \frac{\partial \boldsymbol{x}_i^{(K)}}{\partial \boldsymbol{x}_j} \right\| \leq C_K \left( \sum_{k=0}^{K} \left[ \mathbf{D}^{-1} \left( \mathbf{V} + \mathbf{U} \right) \right]^k \right)_{ij}$$

*where* $C_K$ *is a constant related to the GBN layer. As* $K \to \infty$, $\left| \frac{\partial x_i^K}{\partial x_j} \right|$ *is independent of the distance* $K$.

*Proof.* We give the proof under the scalar-valued case, the vector-valued case can be proved similarly. Denote $T_{ij} = \left[ \mathbf{D}^{-1} \left( \mathbf{V} + \mathbf{U} \right) \right]_{ij}$ and $b_i = \left( \mathbf{D}^{-1} \boldsymbol{\Gamma} \right)_i$, then Eq. (18) can be represented element-wise as

$$x_i^{(k+1)} = \sigma \left( \sum_{j \sim i} T_{ij} \varphi^{(k+1)} \left( x_j^{(k)} \right) \right) + b_i \left( x_i^{(k)} \right) \tag{86}$$

for $k \geq 0$. Now we prove the conclusion

$$\left| \frac{\partial x_i^{(k)}}{\partial x_j} \right| \leq C_k (\delta_{ij} + T_{ij} + \sum_{m=2}^{k} \sum_{j_m,\dots,j_2} T_{ij_m} T_{j_m j_{m-1}} \dots T_{j_2 j}) \tag{87}$$

by induction. For $k = 1$, we have

$$x_i^{(1)} = \sigma \left( \sum_{j \sim i} T_{ij} \varphi^{(1)} \left( x_j \right) \right) + b_i \left( x_i \right) \tag{88}$$

The Jacobian is

$$
\begin{aligned}
\left| \frac{\partial x_i^{(1)}}{\partial x_j} \right| &= \left| \sigma' \cdot \left( \frac{\partial T_{ij}}{\partial x_i} \frac{\partial x_i}{\partial x_j} \varphi^{(1)}(x_j) + T_{ij} \varphi'^{(1)}(x_j) \right) + \frac{\partial b_i}{\partial x_i} \frac{\partial x_i}{\partial x_j} \right| \\
&= \left| \sigma' \cdot \left( \frac{\partial T_{ij}}{\partial x_i} \varphi^{(1)}(x_j) \delta_{ij} + T_{ij} \varphi'^{(1)}(x_j) \right) + \frac{\partial b_i}{\partial x_i} \delta_{ij} \right| \\
&\leq \left| \sigma' \cdot \frac{\partial T_{ij}}{\partial x_i} \varphi^{(1)}(x_j) + \frac{\partial b_i}{\partial x_i} \right| \delta_{ij} + \left| \sigma' \cdot \varphi'^{(1)}(x_j) \right| T_{ij} \\
&\leq C_1 (\delta_{ij} + T_{ij}), 
\end{aligned}
\tag{89}
$$

where $C_1 = \max \left( \sup \left| \sigma' \cdot \frac{\partial T_{ij}}{\partial x_i} \varphi^{(1)}(x_j) + \frac{\partial b_i}{\partial x_i} \right|, \sup \left| \sigma' \cdot \varphi'^{(1)}(x_j) \right| \right)$.

Suppose Eq. (87) holds for $k = K$, i.e.,

$$\left| \frac{\partial x_i^{(K)}}{\partial x_j} \right| \leq C_K (\delta_{ij} + T_{ij} + \sum_{m=2}^{K} \sum_{j_m,\dots,j_2} T_{ij_m} T_{j_m j_{m-1}} \dots T_{j_2 j}) \tag{90}$$

Then, for $k = K + 1$, by the chain rule, we have

$$\left| \frac{\partial x_i^{(K+1)}}{\partial x_j} \right| = \left| \sum_{j_{K+1}} \frac{\partial x_i^{(K+1)}}{\partial x_{j_{K+1}}^{(K)}} \frac{\partial x_{j_{K+1}}^{(K)}}{\partial x_j} \right| \leq \sum_{j_{K+1}} \left| \frac{\partial x_i^{(K+1)}}{\partial x_{j_{K+1}}^{(K)}} \right| \left| \frac{\partial x_{j_{K+1}}^{(K)}}{\partial x_j} \right| \tag{91}$$

$$\left| \frac{\partial x_i^{(K+1)}}{\partial x_{j_{K+1}}^{(K)}} \right| = \left| \sigma' \cdot \left( \frac{\partial T_{ij_{K+1}}}{\partial x_i^{(K)}} \frac{\partial x_i^{(K)}}{\partial x_{j_{K+1}}^{(K)}} \varphi^{(K+1)}(x_{j_{K+1}}^{(K)}) + T_{ij_{K+1}} \varphi'^{(K+1)}(x_{j_{K+1}}^{(K)}) \right) + \frac{\partial b_i}{\partial x_i^{(K)}} \frac{\partial x_i^{(K)}}{\partial x_{j_{K+1}}^{(K)}} \right|$$

$$= \left| \sigma' \cdot \left( \frac{\partial T_{ij_{K+1}}}{\partial x_i^{(K)}} \varphi^{(K+1)}(x_{j_{K+1}}^{(K)}) \delta_{ij_{K+1}} + T_{ij_{K+1}} \varphi'^{(K+1)}(x_{j_{K+1}}^{(K)}) \right) + \frac{\partial b_i}{\partial x_i^{(K)}} \delta_{ij_{K+1}} \right|$$

$$\leq \left| \sigma' \cdot \frac{\partial T_{ij_{K+1}}}{\partial x_i^{(K)}} \varphi^{(K+1)}(x_{j_{K+1}}^{(K)}) + \frac{\partial b_i}{\partial x_i^{(K)}} \right| \delta_{ij_{K+1}} + \left| \sigma' \cdot \varphi'^{(K+1)}(x_{j_{K+1}}^{(K)}) \right| T_{ij_{K+1}}$$

$$\leq C_{K+1,K}(\delta_{ij_{K+1}} + T_{ij_{K+1}}), \tag{92}$$

where $C_{K+1,K} = \max \left( \sup \left| \sigma' \cdot \frac{\partial T_{ij_{K+1}}}{\partial x_i^{(K)}} \varphi^{(K+1)}(x_{j_{K+1}}^{(K)}) + \frac{\partial b_i}{\partial x_i^{(K)}} \right|, \sup \left| \sigma' \cdot \varphi'^{(K+1)}(x_{j_{K+1}}^{(K)}) \right| \right).$

$$\left| \frac{\partial x_{j_{K+1}}^{(K)}}{\partial x_j} \right| \leq C_K (\delta_{j_{K+1}j} + T_{j_{K+1}j} + \sum_{m=2}^{k} \sum_{j_m,\dots,j_2} T_{j_{K+1}j_m} T_{j_m j_{m-1}} \dots T_{j_2 j}) \tag{93}$$

Substituting into Eq. (91),

$$\left| \frac{\partial x_i^{(K+1)}}{\partial x_j} \right| \leq C_K C_{K+1,K} \left[ \delta_{ij_{K+1}} \left( \delta_{j_{K+1}j} + T_{j_{K+1}j} + \sum_{m=2}^{K} \sum_{j_m,\dots,j_2} T_{j_{K+1}j_m} T_{j_m j_{m-1}} \dots T_{j_2 j} \right) \right]$$

$$+ C_K C_{K+1,K} \left[ T_{ij_{K+1}} \left( \delta_{j_{K+1}j} + T_{j_{K+1}j} + \sum_{m=2}^{K} \sum_{j_m,\dots,j_2} T_{j_{K+1}j_m} T_{j_m j_{m-1}} \dots T_{j_2 j} \right) \right]$$

$$= C_K C_{K+1,K} (\delta_{ij} + 2T_{ij} + \sum_{m=2}^{K} \sum_{j_m,\dots,j_2} T_{ij_m} T_{j_m j_{m-1}} \dots T_{j_2 j}$$

$$+ \sum_{j_{K+1}} T_{ij_{K+1}} T_{j_{K+1}j} + \sum_{m=2}^{K+1} \sum_{j_m,\dots,j_2} T_{ij_m} T_{j_m j_{m-1}} \dots T_{j_2 j})$$

$$= C_K C_{K+1,K} (\delta_{ij} + 2T_{ij} + 3 \sum_{j_2} T_{ij_2} T_{j_2 j}$$

$$+ 2 \sum_{m=3}^{K} \sum_{j_m,\dots,j_2} T_{ij_m} T_{j_m j_{m-1}} \dots T_{j_2 j} + \sum_{j_{K+1},\dots j_2} T_{ij_m} T_{j_m j_{m-1}} \dots T_{j_2 j})$$

$$= \leq 9 C_K C_{K+1,K} (\delta_{ij} + T_{ij} + \sum_{m=2}^{K+1} \sum_{j_m,\dots,j_2} T_{ij_m} T_{j_m j_{m-1}} \dots T_{j_2 j}). \tag{94}$$

Let $C_{K+1} = 9 C_K C_{K+1,K}$, we finish the induction.

Since $|\frac{\alpha(x_i)}{\beta(x_i)} - \frac{\alpha(x_i)}{\beta(x_i)}| \leq |x_i - x_j|$ is a contraction, and the Laplacian $\mathbf{L}$ is normalized, $\rho(\mathbf{T}) < 1$. From Lemma B.1, as $K \to \infty$,

$$\lim_{K \to \infty} \left| \frac{\partial x_i^{(K)}}{\partial x_j} \right| = \left[ (\mathbf{I} - \mathbf{T})^{-1} \right]_{ij}, \tag{95}$$

where $\mathbf{T} = \mathbf{D}^{-1} (\mathbf{V} + \mathbf{U})$. $\qquad \square$

## B.9  Derivation of GBN formalism

Set Robin condition coefficients $\alpha_i = \alpha(\boldsymbol{x}_i), \beta_i = \beta(\boldsymbol{x}_i), \gamma_i = \gamma(\boldsymbol{x}_i)$, then Robin condition becomes

$$\alpha_i \boldsymbol{x}_i + \beta_i \sum_{j \sim i, j \in S} (\boldsymbol{x}_i - \boldsymbol{x}_j) = \gamma_i \tag{96}$$

for each $v_i \in \partial S$.

Let the normalized graph Laplacian be reordered to a block matrix

$$\mathbf{L} = \begin{bmatrix} \mathbf{L}_{S,S} & \mathbf{L}_{S,\partial S} \\ \mathbf{L}_{\partial S,S} & \mathbf{L}_{\partial S,\partial S} \end{bmatrix}. \tag{97}$$

Then the Robin condition can be converted into matrix form:

$$\text{diag}(\alpha_i)\mathbf{X}_{\partial S} + \text{diag}(\beta_i)\mathbf{L}_{\partial S,S}\mathbf{X}_S = \mathbf{\Gamma}_{\partial S} \tag{98}$$

Consider the steady solution of the nonhomogeneous heat equation. We have

$$\mathbf{L}_{S,S}\mathbf{X}_S + \mathbf{L}_{S,\partial S}\mathbf{X}_{\partial S} = \mathbf{F}_S, \tag{99}$$

which is the form of Poisson's equation. Merging the above equations, we have

$$\begin{bmatrix} \mathbf{L}_{S,S} & \mathbf{L}_{S,\partial S} \\ \text{diag}(\beta_i)\mathbf{L}_{\partial S,S} & \text{diag}(\alpha_i)\mathbf{I}_{\partial S,S} \end{bmatrix} \begin{bmatrix} \mathbf{X}_S \\ \mathbf{X}_{\partial S} \end{bmatrix} = \begin{bmatrix} \mathbf{F}_S \\ \mathbf{\Gamma}_{\partial S} \end{bmatrix} = \mathbf{\Gamma} \tag{100}$$

Now we use the Jacobi method to solve this nonlinear equation. The diagonal matrix is

$$\mathbf{D} = \begin{bmatrix} \mathbf{I}_{S,S} & \mathbf{0}_{S,\partial S} \\ \mathbf{0}_{\partial S,S} & \text{diag}(\alpha_i)\mathbf{I}_{\partial S,S} \end{bmatrix} \tag{101}$$

The lower diagonal and upper diagonal matrices are

$$\mathbf{V} = \begin{bmatrix} \hat{\mathbf{A}}_{S,S}^{low} & \mathbf{0}_{S,\partial S} \\ -\text{diag}(\beta_i)\mathbf{L}_{\partial S,S} & \mathbf{0}_{\partial S,S} \end{bmatrix} \tag{102}$$

$$\mathbf{U} = \begin{bmatrix} \hat{\mathbf{A}}_{S,S}^{up} & -\mathbf{L}_{S,\partial S} \\ \mathbf{0}_{\partial S,S} & \mathbf{0}_{\partial S,S} \end{bmatrix} \tag{103}$$

for the $k$-th iteration,

$$\text{diag}(\alpha_i) = \alpha(\mathbf{X}^{(k)}; \theta_k) \tag{104}$$

$$\text{diag}(\beta_i) = \beta(\mathbf{X}^{(k)}; \omega_k) \tag{105}$$

$$\mathbf{\Gamma} = \gamma(\mathbf{X}^{(k)}; \eta_k) \tag{106}$$

$$\mathbf{X}^{(k+1)} = \mathbf{D}^{-1}(\mathbf{V} + \mathbf{U})\mathbf{X}^{(k)} + \mathbf{D}^{-1}\mathbf{\Gamma} \tag{107}$$

The $(i, j)$-entry of $\mathbf{D}^{-1}\mathbf{V}$ is

$$(\mathbf{D}^{-1}\mathbf{V})_{ij} = \begin{cases} \frac{1}{\sqrt{d_i^S d_j^S}} & , i \sim j, i \in S, j \in S \\ 0 & , i \sim j, i \in \partial S, j \in \partial S \\ \frac{\beta_i}{\alpha_i \sqrt{d_i^{\partial S} d_j^S}} & , i \sim j, i \in \partial S, j \in S \\ 0 & , else \end{cases} \tag{108}$$

The $(i, j)$-entry of $\mathbf{D}^{-1}\mathbf{U}$ is

$$(\mathbf{D}^{-1}\mathbf{U})_{ij} = \begin{cases} \frac{1}{\sqrt{d_i^S d_j^S}} & , i \sim j, i \in S, j \in S \\ 0 & , i \sim j, i \in \partial S, j \in \partial S \\ \frac{1}{\sqrt{d_i^S d_j^{\partial S}}} & , i \sim j, i \in S, j \in \partial S \\ 0 & , else \end{cases} \tag{109}$$

To give a unified formula, we define the indicator $I_i := \mathbb{I}(i \in S)$, then we define

$$\hat{d}_i = \sum_{j \sim i} I_i I_j + (1 - I_i)(1 - I_j) = d_i(1 - I_i) + (2I_i - 1)\sum_{j \sim i} I_j \tag{110}$$

Then,

$$[(\mathbf{D}^{-1}\mathbf{U})\mathbf{X}^{(k)}]_i = \frac{I_i}{\sqrt{\hat{d}_i}} \sum_{j \sim i} \frac{1}{\sqrt{\hat{d}_j}} \boldsymbol{x}_j^{(k)} \tag{111}$$

$$[(\mathbf{D}^{-1}\mathbf{V})\mathbf{X}^{(k)}]_i = \frac{p_i + (1-p_i)I_i}{\sqrt{\hat{d}_i}} \sum_{j \sim i} \frac{I_j}{\sqrt{\hat{d}_j}} \boldsymbol{x}_j^{(k)}, \tag{112}$$

where $p_i = \frac{\beta_i}{\alpha_i}$. Since $[\mathbf{D}^{-1}\boldsymbol{\Gamma}]_i = \boldsymbol{\Gamma}_i/\alpha_i$, we can replace it with the new variable $\boldsymbol{\Gamma}_i/\alpha_i = \gamma_i$.

Finally we get

$$x_i^{(k+1)} = \frac{I_i}{\sqrt{\hat{d}_i}} \sum_{j \sim i} \frac{1}{\sqrt{\hat{d}_j}} \boldsymbol{x}_j^{(k)} + \frac{p_i + (1-p_i)I_i}{\sqrt{\hat{d}_i}} \sum_{j \sim i} \frac{I_j}{\sqrt{\hat{d}_j}} \boldsymbol{x}_j^{(k)} + \gamma_i \tag{113}$$

## C Algorithm

---
**Algorithm 1** Graph Boundary conditioned message passing Neural network (GBN)

---
**Input:** Graph $\mathcal{G}(\mathcal{V}, \mathcal{E}, \mathbf{A})$, Objective function $\mathcal{J}$, Number of layers $K$, The activation function $\sigma$.
  MLPs with learnable parameters $\alpha(\cdot; \theta_\alpha), \beta(\cdot; \theta_\beta), \gamma(\cdot; \theta_\gamma), \varsigma(\cdot, \eta), \varphi(\cdot, \omega_k)$.
**Output:** Node representations $\mathbf{X}^{(K)}$, Model parameters.
 1: **while** not converge **do**
 2:    Input initial value $\mathbf{X}^{(0)} = \varphi(\mathbf{X}, \omega_0)$;
 3:    Compute soft weights for being boundary node by $I_i = \varsigma(\boldsymbol{x}_i, \eta)$;
 4:    **for** each layer $k = 1$ to $K$ **do**
 5:       Compute coefficients $\alpha_i = \alpha(\boldsymbol{x}_i^{(k-1)}; \theta_\alpha), \beta_i = \beta(\boldsymbol{x}_i^{(k-1)}; \theta_\beta), \gamma_i = \gamma(\boldsymbol{x}_i^{(k-1)}; \theta_\gamma)$;
 6:       Update node representations $\mathbf{X}^{(k)}$ by Eq. (18);
 7:    **end for**
 8:    Compute the objective function $\mathcal{J}(\mathbf{X}^{(K)}, \mathbf{Y})$;
 9:    Train model parameters by Adam optimizer;
10: **end while**

---

**Computational Complexity**   Computing boundary condition coefficients costs $\mathcal{O}(|\mathcal{V}|d)$, where $d$ is the hidden layer dimension. The matrix multiplication is implemented as neighborhood aggregation of complexity $\mathcal{O}(|\mathcal{E}|)$. The overall complexity of a $K$-layer GBN is yielded as $\mathcal{O}(K(|\mathcal{V}| + |\mathcal{E}|))$.

## D Datasets and Baselines

### D.1 Datasets

We evaluate our method on a range of benchmark datasets spanning diverse domains and structural characteristic. Summary statistics are shown in Table 6.

**WikiCS**   A citation graph of Wikipedia articles on Computer Science. Nodes are articles with averaged word embeddings as features; edges represent hyperlinks. The goal is to classify articles into 10 subfields.

**Amazon-Computers**   A product co-purchase graph focused on the "Computers" category from Amazon. Nodes represent products, edges indicate co-purchases, and features are bag-of-words from reviews. The task is product category classification.

**Coauthor-CS**   A co-authorship network from Microsoft Academic Graph. Nodes are authors, edges indicate collaborations, and features are derived from publication texts. The objective is to classify authors into research fields.

**Texas**   A WebKB dataset based on webpages from the University of Texas. Nodes are webpages, edges are hyperlinks, and features come from bag-of-words. The task is to classify webpages into categories like faculty, course, or student.

**Wisconsin**    Similar to Texas, but based on webpages from the University of Wisconsin. Nodes are webpages, edges are hyperlinks, and features use bag-of-words. It is commonly used for evaluating models under small-scale settings.

**Roman-Empire**    A word-level dependency graph from the "Roman Empire" Wikipedia article. Nodes are words; edges connect sequential or syntactically related words. The task is to predict the syntactic role of each word in context. The graph exhibits strong heterophily.

**Amazon-Ratings**    A bipartite graph of users and products from Amazon reviews. Edges represent ratings, with weights as scores. Node features are extracted from review texts. The task is to predict product categories or user preferences.

Table 6: Dataset statistics. The reported number of edges refers to directed edges; for undirected graphs, this count is twice the actual number of edges.

| Dataset | Nodes | Edges | Average Degree | Node Features | Classes | Metric |
|---|---|---|---|---|---|---|
| WikiCS | 11,701 | 431,726 | 36.90 | 300 | 10 | Accuracy |
| Amazon-Computers | 13,381 | 491,722 | 35.76 | 767 | 10 | Accuracy |
| Coauthor CS | 18,333 | 163,788 | 8.93 | 6,805 | 15 | Accuracy |
| Texas | 183 | 650 | 3.55 | 1,703 | 5 | Accuracy |
| Wisconsin | 251 | 1030 | 4.10 | 1,703 | 5 | Accuracy |
| Roman-Empire | 22,662 | 65,854 | 2.91 | 300 | 18 | Accuracy |
| Amazon-Ratings | 24,492 | 186100 | 7.60 | 300 | 5 | Accuracy |

## D.2    Baselines

- **GCN** [33] employs neighborhood aggregation within the spectral domain.

- **GAT** [66] incorporates the attention mechanism into graph-based learning to dynamically weigh neighbor nodes.

- **GIN** [73] enhances graph learning by leveraging MLPs to achieve maximum discriminative power, mimicking the Weisfeiler-Lehman graph isomorphism test process.

- **DR** [26] applies Delaunay triangulation to the node features to rewire the graph, creating a new topology that alters the structural properties to mitigate over-squashing and over-smoothing in graph neural networks.

- **GIN+graphv2** [47] involves processing graph pairs using Graph Isomorphism Networks (GIN) without normalization, focusing on node and graph classification tasks. This approach applies GIN in a batch setting to evaluate performance on various graph datasets.

- **UniFilter** [27] combines low-pass and high-pass filters within each layer and adaptively integrates their embeddings. It also trains a coefficient matrix to measure node correlations for global aggregation.

- **ProxyGap** [28] uses spectral graph pruning to eliminate edges causing over-squashing and over-smoothing, adjusting the graph structure for enhanced neural network performance.

- **Borf** [39] applies Ollivier-Ricci curvature to identify and mitigate over-smoothing and over-squashing in graph neural networks by analyzing and rewriting the graph edges based on curvature values.

- **$G^2$-GCN** [46] introduces gradient gating to dynamically adjust the contribution of each node's gradient during training, allowing different parts of the graph to learn at varying speeds.

- **GREAD** [15] models graph neural networks using a reaction-diffusion process, where node features evolve through diffusion and reaction steps to capture complex graph structures and dynamics.

- **SWAN** [24] examines oversquashing in graph neural networks through the lens of dynamical systems, analyzing how information flow and node interactions affect the network's ability to propagate information effectively.

- **CoBFormer** [70] addresses the over-globalizing problem in graph transformers by selectively focusing on local structures and reducing the emphasis on global information, thereby improving the model's ability to capture fine-grained details.

- **VCR-Graphormer** [21] introduces virtual connections to enable mini-batch processing in graph transformers, allowing for efficient training on large graphs by simulating connections that facilitate information exchange between nodes.

- **Spexphormer** [49] proposes a technique to make graph transformers sparser by selectively retaining important connections and pruning less significant ones, aiming to improve computational efficiency without losing essential information.

- **HGCN** [10] generalizes GAT in the Lorentz model of hyperbolic space in which the graph convolution is conducted in the tangent space. '

- **Graph-mamba** [67] focuses on long-range graph sequence modeling by employing selective state spaces to capture distant dependencies and interactions within graph sequences, enhancing the model's ability to handle complex temporal dynamics.

- **MPNN+VN** [50] investigates the impact of virtual nodes on oversquashing and node heterogeneity in graph neural networks, analyzing how virtual nodes can alter information flow and representation learning.

# E    Implementation Notes

## E.1    Graph transfer

We construct graph transfer datasets inspired by (Gravina et al. 2025), focusing on three types of graph structures: **Line**, **Ring**, and **Crossed-Ring**. Each graph within a task shares the same topology but differs in node features. Specifically, we initialize node features by sampling from a uniform distribution in the interval $[0, 1)$. The source node is initialized with a value of 1, while the target nodes are assigned a value of 0. The objective is to *transfer information from the source node to the target node*, without considering the intermediate nodes. Each graph topology introduces distinct structural properties:

- **Line:** A simple path graph of length $n$, where the source and target nodes are placed at opposite ends, resulting in a shortest path of length $n$.

- **Ring:** Cycles of size $n$, where the source and target nodes are placed at a distance of $\lfloor n/2 \rfloor$ from each other.

- **Crossed-Ring:** Cycles with additional "cross" edges between intermediate nodes. These edges do not reduce the shortest path between source and target, which remains $\lfloor n/2 \rfloor$.

Figure 7 illustrates each graph structure with a source–target distance of $n = 5$. In our experiments, we consider distances $n \in \{3, 5, 10, 50\}$, and use message passing neural networks (MPNNs) with depth equal to $n$. Unless stated otherwise, input dimension is set to 1 (regression tasks), hidden dimension to 64, and training is run for 2000 epochs. We apply node masking during training and testing to compute loss solely based on the source and the target node. We generate 1000 graphs for training, 100 for validation, and 100 for testing. Final performance is reported as the mean squared error over the test set. The architectural details and hyperparameters of our model are summarized in Table 7.

Table 7: Hyperparameter settings for graph transfer task.

| Dataset | hid_dim | Activation | dropout | norm | lr | w_decay |
|---|---|---|---|---|---|---|
| Line | 64 | Tanh | 0 | BatchNorm | 1e-3 | 0 |
| Ring | 64 | Tanh | 0 | BatchNorm | 1e-3 | 0 |
| Crossed-Ring | 64 | Tanh | 0 | BatchNorm | 1e-3 | 0 |

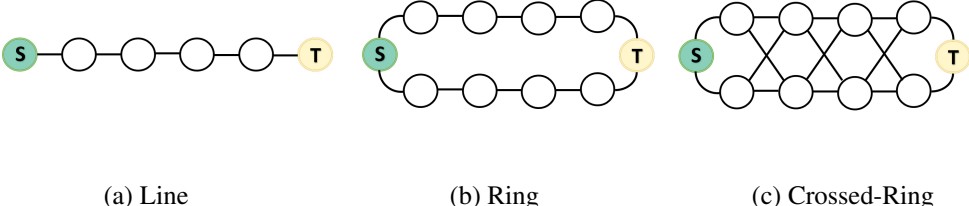

|                   |                  |                         |
| :---------------: | :--------------: | :---------------------: |
| (a) Line          | (b) Ring         | (c) Crossed-Ring        |

Figure 7: In all three graphs—Line, Ring, and Crossed-Ring—the source node "S" and target node "T" are placed five hops apart.

## E.2 Further Details on Visualization

We visualize the graph transfer task in Fig. 8 to investigate the model performance, comparing the vanilla GCN and the proposed GBN. To be specific, the experiment is conducted on the graph where two 10-node complete graphs (i.e., the source on the left numbered $14$-$23$ and the target on the right $0$-$9$) are connected by a line numbered $10$-$13$ (i.e., topological bottleneck). The nodes are randomly assigned in $[0, 1]$ in the source, while $[-1, 0]$ in the target. The node value in the connecting line is "$0$". The task is to swap the values in the source and target components. As a result, GCN results in the messages being severely "squashed" in the bottleneck, as shown in the nodes numbered $9$, $10$, $13$ and $14$. In contrast, the proposed GBN allows messages to pass through the bottleneck, thanks to local bottleneck adjustment.

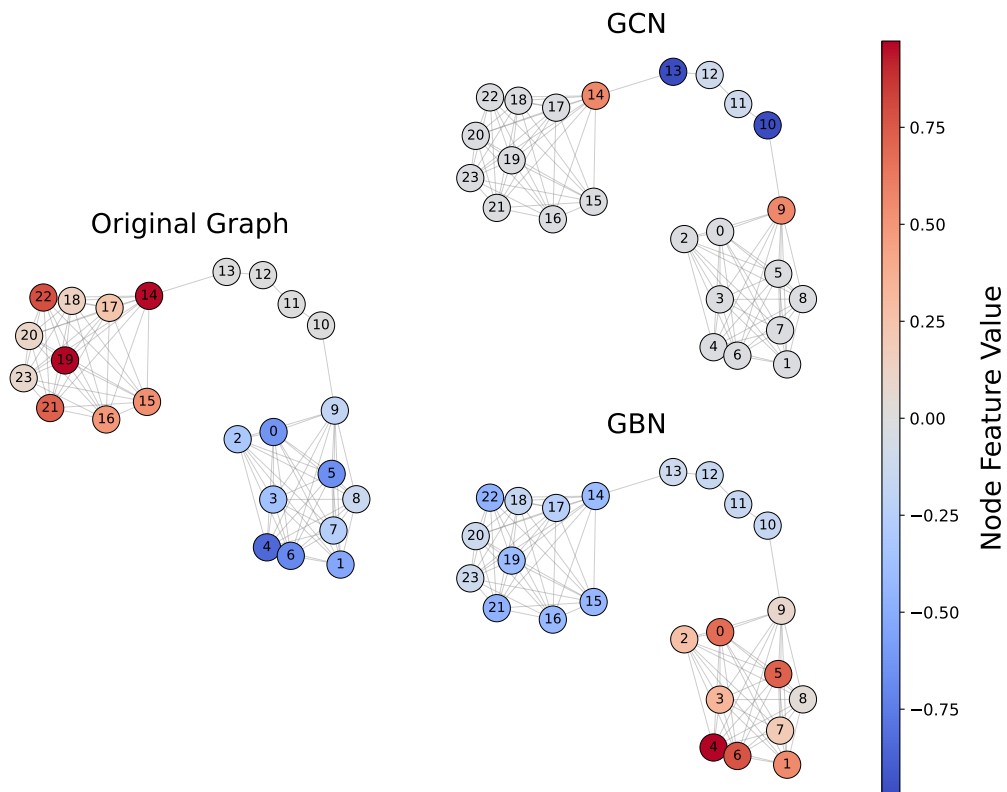

Figure 8: Case Study on Graph Transfer

### E.3 Hyperparameter settings

We conduct node classification experiments using a 2-layer GBN network, all the datasets are available as PyTorch Geometric datasets. All datasets are split into 50% training, 25% validation, and 25% testing. We report the final test accuracy as the average over 10 random data splits. The same experimental setup is applied across baseline methods. For experiments on heterogeneous graphs, we utilize a 5-layer model. The architectural details and hyperparameters are summarized in Table 8. Learning rates are set to 3e-5, with a dropout rate to 0.3, 0.2, 0.1 and a hidden dimension size of 512. LayerNorm is used to facilitate deeper models. All experiments are implemented using PyTorch Geometric library. The hardware is NVIDIA GeForce RTX 4090 GPU 24GB memory, and Intel Xeon Platinum 8352V CPU with 120GB RAM. Our code is publicly available at `https://anonymous.4open.science/r/GBN-E854`:

Table 8: Hyperparameter settings for node classification task.

| Dataset | n_layers | hid_dim | Activation | dropout | norm | lr | w_decay |
|---------|----------|---------|------------|---------|------|-----|---------|
| Texas | 5 | 512 | GELU | 0.3 | LayerNorm | 1e-4 | 0 |
| Wisconsin | 5 | 512 | GELU | 0.7 | LayerNorm | 1e-3 | 0 |
| Amazon-Ratings | 5 | 512 | GELU | 0.15 | BatchNorm | 3e-4 | 0 |
| Roman-Empire | 5 | 512 | GELU | 0.15 | LayerNorm | 3e-4 | 0 |
| Coauthor-CS | 3 | 512 | GELU | 0.2 | LayerNorm | 3e-5 | 0 |
| AmazonComputers | 3 | 512 | GELU | 0.2 | LayerNorm | 3e-5 | 0 |
| WikiCS | 3 | 512 | GELU | 0.2 | LayerNorm | 3e-5 | 0 |

## F  Broader Impact and Limitations

**We pave the way to design deeper and better expressive MPNNs to learn on more sophisticated graphs, offering new potentials of graph foundation models.** A positive societal impact lies in the scalability of our design, allowing for the analysis on large scale real-world graphs. None of negative societal impacts we feel must be specifically highlighted. As for limitations, our work as well as typical MPNNs primarily considers the undirected graphs, while the message passing on directed graphs has left to be the future work.

