# OpenReview forum: "Deeper with Riemannian Geometry: Overcoming Oversmoothing and Oversquashing for Graph Foundation Models"
_NeurIPS.cc/2025/Conference — NeurIPS 2025 poster_

### Official Review · Reviewer_sXZ5 · 2025-06-15

**Clarity:** 3
**Significance:** 4
**Originality:** 4
**Rating:** 6
**Confidence:** 4

**Summary:**

In this submission, the authors propose to mitigate both oversmoothing and oversquashing issues of MPNNs. From the lens of Riemannian geometry, this paper begins with showing the increase of lambda leading to gradient vanishing, and thus present a message-passing paradigm of local adjustment, leveraging the proposed nonhomogeneous boundary condition. Consequently, a provable GBN is designed with the Robin condition, and is evaluated on both homophilic and heterophilic graphs.

**Questions:**

Q1. Link prediction performance comparison with GCN, GAT, GIN and SAGE on typical benchmark datasets such as Cora, Citeseer and CS.

Q2. Does GBN need to explicitly identify the boundary nodes? Does there exist graphs having no boundary node, or to what extent? Under that case, GBN degenerates to GCN exactly?

Q3. Please specify the computational overhead again the vanilla GCN.

Q4. Implementation details, i.e., the initial values of boundary condition coefficients, and the neural network architecture for the indicator I.

**Ethical Concerns:**

["NO or VERY MINOR ethics concerns only"]

**Final Justification:**

After reading the authors’ responses and other reviewers’ comments, I confirm that the proposed GBN mitigates both oversmoothing and oversquashing while maintaining the computational complexity of conventional GCN yet not altering the original topology.

Thus, this submission shows a new promising direction on MPNN. I vote for the acceptance and raise my scores accordingly.

**Limitations:**

yes

**Quality:**

4

**Strengths And Weaknesses:**

Strengths

S1. The studied problem is significant given oversmoothing and oversquashing are longstanding issues in MPNNs. The authors theoretically discover the possible negative impact of globally increasing the spectral gap, and it is interesting to study the local adjustment MPNN alternatively.

S2. The established theory is solid. The authors unify the MPNN formalism from Riemannian dynamics (manifold heat kernel) and, in order to jointly address the both issues, construct a nonhomogeneous boundary condition on the manifold with rigorous elaboration. No error is detected in the proof.

S3. The proposed GBN is novel and its complexity is the same as vanilla GCN. The authors design a new message passing paradigm based on nonhomogeneous Robin condition with theoretical guarantees.

S4. The experiments on homophilic and heterophilic graphs are convincing. The authors evaluate the ability of GBN against oversmoothing and oversquashing, and show model performance in very deep network. Codes are available.

Weaknesses

W1. The authors give the time complexity comparison with some baselines, while GraphNormv2, G2-GCN and SWAN are not listed.

W2. Further details should be provided in the paper, e.g., is Lemma 5.3 only gives an existence about the solution of over-smoothing? See Q3. The theory in this paper is not friendly to the readers not familiar with Riemannian dynamics and PDEs.

---

> ### Author Rebuttal · Authors · 2025-07-30
>
> ### **1. Time complexity comparison**
> |Model|Complexity|
> |---|---|
> |GraphNormv2|$O(L\cdot(\lvert \mathcal{V} \rvert + \lvert \mathcal{E} \rvert))$|
> |G2-GCN|$O(T\cdot(\lvert \mathcal{V} \rvert + \lvert \mathcal{E} \rvert))$|
> |SWAN|$O(T\cdot(\lvert \mathcal{V} \rvert + \lvert \mathcal{E} \rvert))$|
> |GBN|$O(L\cdot(\lvert \mathcal{V} \rvert + \lvert \mathcal{E} \rvert))$|
>
> $T$ is the steps of ODE, $L$ is the number of layers. GBN has the same time complexity as GraphNormv2.
>
> ### **2. On Lemma 5.3**
> Lemma 5.3 claims that there exists some source term to combat over-smoothing, so that we can learn such source term with an MLP, supporting our design.
>
> ### **3. Link prediction performance comparison to traditional MPNNs**
> We show the link prediction results in terms of AUC (\%).
>
> |Model|Cora|Citeseer|CS|
> |---|---|---|---|
> |GCN|90.11(0.51)|90.16(0.49)|92.71(0.63)|
> |GAT|92.55(0.49)|90.32(0.36)|92.80(0.75)|
> |SAGE|90.02(0.55)|91.18(0.22)|92.44(0.76)|
> |GIN|91.02(0.37)|90.06(0.37)|91.58(0.46)|
> |GBN|97.93(0.44)|98.26(0.56)|98.21(0.34)|
>
> The proposed GBN consistently achieves the best link prediction results on Cora, Citeseer and CS datasets, and receives at least 5\% performance gain compared to traditional MPNNs.
>
> ### **4. Boundary nodes and the connection to GCN**
> We specify that the proposed GBN does not need to explicit identify the boundary nodes, but it is equipped with a neural boundary indicator adopting the soft assignment. That is, the boundary nodes are jointly learned with the model.
>
> There exist special cases of graphs without boundary nodes, e.g., a complete graph where each node is connected to all the other nodes. The soft indicator approaches to $0$ for all the nodes.
>
> We emphasize that GBN is applicable to any graphs, regardless of the existence of boundary nodes, since we locally adjust the message-passing at the boundary nodes with soft assignment.
> Also, the proposed GBN recovers the vanilla GCN when $\gamma$ and $\beta$ are eliminated.
>
> ### **5. Computational overhead against the vanilla GCN**
> The computational complexity of the proposed GBN is $\mathcal O(\|\mathcal V\|+\|\mathcal E\|)$, which is the same as the vanilla GCN, where $\mathcal O(\|\mathcal V\|)$ is for feature transformation and $\mathcal O(\|\mathcal E\|)$ for aggregation. $\mathcal V$ and $\mathcal E$ denote the node set and edge set, respectively.
>
> That is, we jointly mitigate over-smoothing and over-squashing via a new message-passing paradigm grounded on Riemannian boundary condition with no computational overhead.
>
> ### **6. Implementation details**
> 1. For the initial values of boundary condition coefficients, $\gamma$ and $\beta$ are given as 2-layer MLPs. We do not need to explicitly construct $\alpha$, since $\alpha$ is nonzero, and can be divided by both $\gamma$ and $\beta$. Thus, $\gamma$ and $\beta$ are randomly initialized and jointly learned with the model.
>
> 2. For the neural architecture of the indicator, the indicator is a learnable function to determine whether or not the node is on the boundary, and it needs the neighborhood information for the boundary judgment. Accordingly, we conduct feature mapping and aggregation. Then, a sigmoid function is utilized to get the soft assignment values to identify the boundary nodes. This indicator only computes once in the beginning.

---

> > ### Comment · Reviewer_sXZ5 · 2025-08-06
> >
> > After reading the authors’ responses and other reviewers’ comments, I confirm that the proposed GBN mitigates both oversmoothing and oversquashing while maintaining the computational complexity of conventional GCN yet not altering the original topology. Thus, this submission shows a new promising direction on MPNN. I vote for the acceptance and raise my scores accordingly.

---

> > > ### Author Response · Authors · 2025-08-06
> > >
> > > Thanks again for your constructive comments!
> > >
> > > We are so glad to address your concerns and receive your appreciation!

---

### Official Review · Reviewer_QJWv · 2025-06-24

**Clarity:** 2
**Significance:** 2
**Originality:** 3
**Rating:** 4
**Confidence:** 4

**Summary:**

The paper identifies that existing global remedies for oversquashing in MPNNs, specifically rewiring approaches that enlarge the graph's spectral gap, actually accelerate gradient vanishing and oversmoothing. For this, they draw on the continuous analogue of studying heat flow on Riemannian manifolds. To overcome these issues, they introduce *Graph Boundary-conditioned MPNNs* which adapt the message passing to include a learnable nonhomogenous Robin boundary condition, which can adaptively regulate information flow and mitigate both issues. They demonstrate the theoretical benefits of such a boundary condition for the continuous analogue, e.g., by analyzing Dirichlet energy decay. Empirically, the authors show that GBN outperforms a large set of baselines on standard small to medium-scale homophilic/heterophilic benchmarks, scales beyond 256 layers with no performance drop on Cora, and effectively mitigates oversquashing measured by performance on the standard graph transfer task.

**Questions:**

- Overall presentation and clarity of the paper should be improved, mainly self-containedness of § 6 (see sec. on Weaknesses), but also the general writing (just for example, the term "GBN" is introduced without explanation in the abstract, or the title of § 6 is "GBN: A Provable Neural Architecture").
- Evaluation on the benchmarks from the Weaknesses section would make a stronger case for GBN.
- The paper could dedicate more space to providing some intuition for GBN. After all, as far as I understand Eq (18) also implements some shift operator wrt. a weighted graph, whose filter response could be analyzed. Also, it would be valuable for intuition to, e.g., plot the boundary assignment $I_i = \zeta(x_i, \eta)$ for some specific graphs.
- The related work section could mention some more recent approaches to achieve similar goals (such as, e.g., [2]).

[2] *Unitary convolutions for learning on graphs and groups.* Bobak T. Kiani, Lukas Fesser, Melanie Weber. NeurIPS 2024.

**Ethical Concerns:**

["NO or VERY MINOR ethics concerns only"]

**Final Justification:**

The authors resolved most of my concerns with the paper. I believe the method mitigating oversquashing without altering the graph's topology is conceptually nice and can be valuable to the community. I am confident that the inclusion of larger-scale and long range benchmarks, and an improved presentation of § 6, will improve the paper's impact. While I still believe that the lengthy motivation on manifolds is not very accessible to a graph learning audience (and maybe not necessary to understand the method), I will recommend acceptance.

**Limitations:**

yes

**Paper Formatting Concerns:**

The main body seems to make excessive use of reduced spacing between sections/subsections. While I didn't find anything in the 2025 style guidelines explicitly prohibiting this, such adjustments may impact readability and visual consistency with other papers. To my knowledge, similar deviations have sometimes been considered violations of submission policies in the past. I kindly ask the AC to check whether this is okay, and ask the authors to adjust it if not.

**Quality:**

3

**Strengths And Weaknesses:**

**Strengths:**
- To the best of my knowledge, the idea of considering such adaptive boundary conditions is novel and, in my view, a quite elegant way compared to the "static" graph rewiring approaches that have been proposed.
- The method seems to introduce no significant computational overhead (i.e., scales similarly to GCNs).
- Strong empirical performance for node classification, and the claims of mitigating oversmoothing and oversquashing are effectively verified as there is no performance degradation even with 256 layers and Dirichlet energy stays roughly constant.

**Weaknesses:**
- The main body leans heavily on Riemannian geometry (§ 5) to motivate GBN, but most of the benchmark graphs lack any clear manifold interpretation which makes the link seem somewhat loose. The resulting core GNN update in § 6 is barely discussed in the main body. Also, the fact that the boundary set is implemented dynamically via a learned, soft gate is only vaguely mentioned in the main body at lines 247-248 and only looking at the appendix/code helped clear things up for me. I understand that this might be a natural thing to do in PDE circles, but I would suspect that a GNN-focused audience might benefit from a more self-contained presentation.
- For claiming scalability, the experiments are quite small-scale (and for the oversquashing experiment only on synthetic data). To validate (a) scalability, it could make sense to evaluate on `ogbn-{arxiv, products}`. For (b) long range dependencies, it might make sense to also evaluate on some tasks of the long range graph benchmark (such as `peptides-{func, struct}`) [1].

[1] *Long Range Graph Benchmark.* Vijay Prakash Dwivedi, Ladislav Rampášek, Mikhail Galkin, Ali Parviz, Guy Wolf, Anh Tuan Luu, Dominique Beaini. NeurIPS 2022.

---

> ### Author Rebuttal · Authors · 2025-07-30
>
> ### **1. Manifold interpretation of benchmark graphs**
> The benchmark graphs are inherently connected to a Riemannian manifold. Theoretically, a graph is regarded as the discrete analogy to a certain Riemannian manifold [1]. It motivates us to study MPNN with Riemannian geometry. As an empirical observation, we take the Computer dataset for instance. Rather than Euclidean, it can be decomposed as a -1.15-curvature hyperbolic component of weight 0.61, a 0-curvature component of weight 0.18, and a +0.72-curvature hyperbolic component of weight 0.21.
>
> Note that we do not focus on the optimal embedding space or curvature distribution of the graphs, but explore the functional space over the manifold to combat over-smoothing and over-squashing.
>
> [1] Chung, F.R.K. Spectral Graph Theory. AMS, 1997.
>
> ### **2. The term of GBN**
> We will improve the presentation. The term comes from Graph Boundary Conditioned Message Passing Neural Network as in Line 228. We will explain GBN in Abstract in Line 13 as follows:
> “We design a Graph Boundary conditioned message passing Neural network (GBN) with local bottleneck adjustment”
>
> ### **3. Scalability on large graphs and long-range graph benchmark**
> For large graphs, we do the experiment on NVIDIA GeForce RTX 4090, and 16 vCPU Intel Xeon Gold 6430.
>
> |Datasets|Metric|GCN|ProxyGap| MPNN+VN| $\text{G}^2\text{-GCN}$|SWAN|PANDAS|GBN|
> |---|---|---|---|---|---|---|---|---|
> |ogbn-arixv|ACC| 71.53 ± 0.22| 72.02±0.76|OOM| 73.45±0.76|73.26±1.80 | 72.32±0.46|73.85±0.46|
> |ogbn-product|ACC| 80.33±0.19|OOM | OOM| 81.23±0.51| 80.98±0.31| OOM|82.40±0.32|
> |peptides-func|AP| 59.13±0.17| 65.64±0.58|66.18±0.67|62.73±0.43 |64.86±0.49 | 60.14±0.34|65.88±0.73 |
>
>
> OOM is short for Out-Of-Memory. We achieve the best node classification results on ogbn-arXiv and ogbn-product, and promising graph-level task performance on peptides-func.
>
> ### **4. Updating rule in Sec. 6**
> We do appreciate the careful review on Appendix/Code for the derivation and implementation of the GBN’s updating rule.
>
> In addition, we would like to provide more explanation of the proposed updating rules:
> 1. We first describe the behavior of the interior nodes and boundary nodes behavior in heat diffusion together. Without loss of generality, we first assume that the boundary nodes are recognized, then by reordering the nodes, we have the blocked Laplacian matrix and equation system as in Eq. (14). Then, the heat diffusion proceeds on the interior subgraphs, influenced by boundary condition, which form a linear equation in matrix form in Eq. (100), Appendix B.9.
> 2. Then we use the numerical method (e.g., Jacobi method) to **iteratively solve the linear equation** in Eq. (100), the details are in Lemma 6.2. The results in Eq. (18) exactly meets **the form the traditional MPNNs** but **has different essential and more favorable properties that prevents over-smoothing and over-squashing.**
> 3. To adopt learning ability, we fix the results in Lemma 6.2 with nonliear activation functions and learnable MLPs, so that we obtain the updating rules Eq. (18) which still have nice properties that prevents over-squashing by Theorem 6.3 and over-smoothing by nonhomogeneous boundary condition in Lemma 5.3 and Theorem 5.4.
> 4. Now, under above framework, we need to determine which or how which a node being a boundary nodes. Since boundary condition affects a lot on its neighborhood, so we adopt a simple GCN layer (only include aggregation and MLP) and a sigmoid function to compute soft weights of indicators in Eq. (17) at the beginning phase of GBN architecture. This allows gradients to flow through the boundary detection mechanism, enabling end-to-end adaptation.
> 5. To make the boundary condition more flexible and be able to learn complex graph structures, we assign different $(\alpha, \beta, \gamma)$ to each boundary node. The values are computed by 2-layer MLPs.
> 6. The model name GBN comes from the important tool, the boundary condition, we use to solve the over-smoothing and over-squashing problem in GNN. The complete name is Graph Neural Network with Boundary Condition.
>
> ### **5. On Eq. 18**
> It is interesting to view Eq. (18) as a graph filter. From the lens of graph signal processing, the matrix $\mathbf{D}^{-1}(\mathbf{V}+\mathbf{U})$ can be treated as a local graph shift operator, which adaptively interpolates between a low-pass filter and a high-pass filter.
> The filter's property is controlled by Robin boundary condition, i.e., each pair $(\alpha_i, \beta_i)$ or $p_i=\frac{\beta_i}{\alpha_i}$:
> - It tends to act as a low-pass filter as $p_i$ increase, while shifting to high-pass filter if $p_i$ approaches 0.
> - For $p_i \in (0,1)$, it is a band-pass filter that chooses intermediate frequencies to flow by.
>
> ### **6. Visualization of Boundary Assignment**
> We use the synthetic graph in the Case Study for elaboration. As shown in Fig. 1 in the supplementary material, the synthetic graph consists of two 10-node complete graphs (i.e., the source on the left of nodes 14-23 and the target on the right of nodes 0-9) with a line connecting them (nodes 10-13).
> Owing to the NeurIPS policy, we present the visualization of boundary assignment as in the table below.
>
> |1|2|3|4|5|6|7|8|9|10|11|12|13|14|15|16|17|18|19|20|21|22|23|24|
> |---|---|---|---|---|---|---|---|---|---|---|---|---|---|---|---|---|---|---|---|---|---|---|---|
> |99.99|99.99|99.99|99.99|99.99|99.99|99.99|99.99 | 100|99.98 |99.90 |98.88 |98.88 |97.87| 13.62| 19.84|19.03 |14.45 |12.95 |30.48 | 12.52|24.06 |25.67 |12.34 |
>
> Note that, the joint node 9 of the line and the right complete graph is labelled as the boundary of 100\%, which lies in the topological bottleneck where the message passing should be locally adjusted. Note that, the nodes in the right complete graph are marked as 99.9\%, since they are topologically equivalent and affected by node 9.
> Also, the nodes 10-13 at the bottleneck need to adjust the information flow, and they are softly assigned as the boundary nodes.
>
> ### **7. Related Work of NeurIPS’24 paper (Unitary convolutions for learning on graphs and groups).**
> We will include the discussion of the NeurIPS’24 paper, which is an interesting work to mitigate over-smoothing with the solid theory of unitary group convolution.
>
> We will cite it in Sec. of Related Work-Over-smoothing as follows:
> ```
> [NeurIPS’24] present a novel idea to transform the global spectral distribution in each layer of MPNN, while we focus on local adjustment of MPNN via boundary conditions.
> ```

---

> > ### Comment · Reviewer_QJWv · 2025-08-04
> > **Response by Reviewer**
> >
> > I thank the authors for their thoughtful rebuttal, which addressed many of my concerns. Here my detailed answers:
> >
> > > **2. The term of GBN**
> > We will improve the presentation. The term comes from Graph Boundary Conditioned Message Passing Neural Network as in Line 228. We will explain GBN in Abstract in Line 13 as follows: “We design a Graph Boundary conditioned message passing Neural network (GBN) with local bottleneck adjustment”
> >
> > If the authors properly introduce the term at its first occurrence, I consider this resolved.
> >
> > > **3. Scalability on large graphs and long-range graph benchmark**
> > For large graphs, we do the experiment on NVIDIA GeForce RTX 4090, and 16 vCPU Intel Xeon Gold 6430.
> >
> > I thank the authors for providing these additional experiments. Personally, I would see the inclusion of the OGBN datasets more as evidence that the method scales favorably. I wouldn't have expected significantly improved performance on these tasks, as to my knowledge they do not suffer significantly from oversmoothing/-squashing. The authors’ results confirm this. Still, I think this practically demonstrates the method's scalability, which can be a nice addition. Regarding the peptides experiment, I am not entirely convinced by the results yet, as the comparison was based on an untuned GCN baseline. A tuned version of GCN with a few more layers can actually achieve much better results [1], so a comparison with this would be more adequate for a potential camera-ready version.
> >
> > > **4. Updating rule in Sec. 6**
> > We do appreciate the careful review on Appendix/Code for the derivation and implementation of the GBN’s updating rule.
> >
> > I thank the authors for the comprehensive explanation. However, my point in the review was that I feel § 6 is too short and some important information on the method (like that the boundary assignment being soft+learned) is hidden in the appendix. I think that the paper would benefit a lot from § 6 having a better, more self-contained presentation.
> >
> > > **6. Visualization of Boundary Assignment**
> > We use the synthetic graph in the Case Study for elaboration. As shown in Fig. 1 in the supplementary material, the synthetic graph consists of two 10-node complete graphs (i.e., the source on the left of nodes 14-23 and the target on the right of nodes 0-9) with a line connecting them (nodes 10-13). Owing to the NeurIPS policy, we present the visualization of boundary assignment as in the table below.
> >
> > I thank the authors for including the table. Including such a visualization will greatly improve intuition.
> >
> > Conditional on the authors improving the explanations in §6 for the camera-ready version, and pending the AC's confirmation that the formatting is acceptable (or the authors addressing it), I will raise my score to **4: Borderline accept**.
> >
> > [1] *Jan Tönshoff, Martin Ritzert, Eran Rosenbluth, Martin Grohe.* Where Did the Gap Go? Reassessing the Long-Range Graph Benchmark. arXiv:2309.00367

---

> ### Author Response · Authors · 2025-08-05
> **Comment by Authors**
>
> Dear Reviewer QJWv,
>
> Thanks again for your time and constructive comments! We are so glad to address your concerns.
>
> First, we would like to specify **the empirical settings** of Long-range Graph Benchmark in the rebuttal.
> GCN follows the original setting as (Long Range Graph Benchmark. Vijay Prakash Dwivedi, Ladislav Rampášek, Mikhail Galkin, Ali Parviz, Guy Wolf, Anh Tuan Luu, Dominique Beaini. NeurIPS 2022), which is untuned.
>
> Next, we show **the performance comparison following [1]**, where Laplacian positional encoding is adopted by both GCN and the proposed GBN. We train GBN with 1400 iterations, and show the results as follows,
> |Datasets|Metric|GCN|tuned-GCN|GBN (1400 iterations)|
> |---|---|---|---|---|
> |peptides-func|AP|59.13±0.17|68.60 ± 0.50|68.47±0.64|
>
> We will specify the term of GBN at its first occurrence, improve the elaboration in Sec. 6, and include the visualization in the one additional page in the camera-ready version.
>
> Also, we are dedicated to completing the experiment of all the baselines with the tuned setups, and including a detailed discussion.
>
> Your endorsement is important to us!
>
> Best regards,
>
> All authors of submission 6779

---

### Official Review · Reviewer_wr1V · 2025-06-27

**Clarity:** 3
**Significance:** 3
**Originality:** 3
**Rating:** 4
**Confidence:** 4

**Summary:**

This paper addresses two fundamental, interconnected problems in MPNNs: oversmoothing and oversquashing.
The authors challenge the prevailing "global" approaches (e.g., graph rewriting) that aim to fix these issues by modifying the entire graph structure, often by increasing the spectral gap. Through a theoretical analysis grounded in Riemannian geometry. As an alternative, the paper proposes a "local" approach that refines the message-passing mechanism itself. By modeling MPNNs as solvers for a heat equation on a Riemannian manifold. This theoretical foundation leads to their proposed model, the Graph Boundary conditioned message passing Neural network (GBN).

**Questions:**

See weaknesses above.

**Ethical Concerns:**

["NO or VERY MINOR ethics concerns only"]

**Final Justification:**

The authors have addressed some of my concerns, but the strength of their contribution remains limited by the fact that local Ricci curvature has already been proposed and applied in prior work (Mitigating Over-Smoothing and Over-Squashing using Augmentations of Forman-Ricci Curvature, LoG 2023). While it is novel to employ this metric within a non-rewiring framework, the core idea is not entirely new. I am maintaining a borderline rating for this paper, though my evaluation has become somewhat more positive.

**Limitations:**

No additional limitations are identified.

**Quality:**

3

**Strengths And Weaknesses:**

### Strengths
1. The paper tackles oversmoothing and oversquashing, which are critical and well-recognized challenges that limit the depth and expressive power of modern GNNs.
2. The authors provide theoretical analysis, connecting MPNNs to heat flow on Riemannian manifolds.
3. GBN offers a promising direction for mitigating oversquashing without modifying the graph's topology. This is a significant advantage over conventional graph-rewiring methods, which can be computationally expensive and risk altering essential structural information.

### Weaknesses
1. The authors claim to "put forward a local treatment and introduce the local Riemannian geometry for the first time." This claim is questionable, as several well-studied methods already leverage local geometric concepts. For instance, methods based on Ollivier-Ricci Curvature [1] and Forman-Ricci Curvature [2] are intrinsically local, as they operate on small neighborhoods. This undermines a central claim of the paper's novelty.

2. The paper fails to compare against some highly relevant baselines. For example, PANDA [3] is another recent attempt to tackle oversquashing without altering the graph topology, making it a critical point of comparison that is missing from the evaluation.
3. The paper's use of "external inputs" to combat oversmoothing appears conceptually similar to well-established techniques like initial residual connections.
4. The paper argues that edge deletion, used by some methods to combat oversmoothing, "hinders the information flow." This may be a mischaracterization of prior work [2, 4]. Methods like Spectral Graph Pruning or those using Ricci curvature typically only prune edges in overly dense regions, which is not necessarily a hindrance to overall information flow and may even be beneficial.
5. The introduction of multiple MLPs for the boundary conditions adds a new set of hyperparameters. The paper does not explore the model's sensitivity to these choices.

> References

[1] Revisiting Over-smoothing and Over-squashing Using Ollivier-Ricci Curvature

[2] Mitigating Over-Smoothing and Over-Squashing using Augmentations of Forman-Ricci Curvature

[3] PANDA: Expanded width-aware message passing beyond rewiring.

[4] Spectral Graph Pruning Against Over-Squashing and Over-Smoothing.

---

> ### Author Rebuttal · Authors · 2025-07-30
>
> ### **1. On novelty and local Riemannian geometry**
> The novelty of our work lies in that we introduce the local adjustment to message passing paradigm from the perspective of Riemannian geometry. The notion of “local” emphasizes that we locally adjust message passing via boundary condition, rather than any global criteria, e.g., spectral gap. Also, we clarify that: our theoretical contribution is not the introduction of geometric notion but the establishment of nonhomogeneous boundary condition.
>
> Ricci curvature is defined on the edges and describes the connectivity over the graph. According to the definition (Eq. 4 in [1]), calculating Ricci curvature necessitates finding shortest paths between all node pairs (or sampled neighborhoods), inherently traversing the entire graph.
>
> We recognize the effectiveness of (curvature-guided) graph rewriting methods, and consider that they are the natural solution as over-squashing is related to the topological bottlenecks in the graph structure. However, our work is inherently different from this research line:
> - Graph rewriting focuses on altering the original graph structure so that over-smoothing/over-squashing is mitigated under traditional message-passing.
> - We maintain the original graph structure but refine the message-passing paradigm to jointly mitigate over-smoothing and over-squashing.
>
> We will paraphrase this sentence for clarity as follows:
> ```
> We introduce the local adjustment to the message passing paradigm with nonhomogeneous boundary conditions of Riemannian geometry.
> ```
>
> ### **2. On ICML’24 (PANDA)**
> Node Classification Performance in terms of ACC on 7 Datasets
>
> |Models|WikiCS| Computer| CS| Texas| Wisconsin|RomanEmpire|Ratings|
> |---|---|---|---|---|---|---|---|
> |PANDA|84.86±0.43|90.18±0.76|93.62±0.44|80.89±0.44|83.98±1.58|86.23±0.66|52.35±0.34|
> |GBN|86.21±0.39|91.33±0.32|95.78±0.21|85.01±6.51|86.78±3.84|89.83±0.46|53.51±0.88|
>
> The proposed GBN consistently achieves better classification results than PANDA.
>
> Performance on Long-range Dependence in terms of MSE
>
> |Models|Line 3|Line 5|Line 10|Line 50|
> |---|---|---|---|---|
> |PANDA|3.13e-7|5.76e-7|8.67e-7|3.71e-6|
> |GBN|2.04e-7|3.76e-7|7.51e-7|3.55e-6|
>
> |Models|ring 3|ring 5|ring 10|ring 50|
> |---|---|---|---|---|
> |PANDA|5.13e-7|6.76e-7|5.67e-6|1.21e-5|
> |GBN|2.6e-7|6.98e-7|2.13e-6|7.73e-6|
>
> |Models|CrossedRing 3|CrossedRing 5|CrossedRing 10|CrossedRing 50|
> |---|---|---|---|---|
> |PANDA|6.13e-7|1.76e-6|5.67e-6|1.21e-5|
> |GBN|4.82e-7|1.76e-6|2.74e-6|5.37e-6|
>
> The proposed GBN presents superior information transfer ability on Line, Ring, and Cross-Ring datasets. It shows the ability to capture long-range dependence.
>
> We will include the discussion of this insightful paper in Sec. of Related Work, and cite it as follows:
> ```
> To combat over-squashing, [PANDA] present a novel approach to select nodes with high centrality to expand in width to encapsulate the growing influx of signals from distant nodes.
> ```
>
> In addition, we would like to specify the difference between PANDA and the proposed GBN:
> PANDA changes the way of neighborhood aggregation without topological considerations, but GBN combines the (topological) boundary condition with functional behavior.
>
> ### **3. Difference between external input and initial residual connection**
> On the formulation:
> The domain of definition is different. Concretely, the initial residual connection is uniformly applied to all nodes. In contrast, the proposed external input (the $\gamma$-term) is exclusively defined for boundary nodes as in Definition 6.1 and Eq. 14.
> The formulation of external input in Eq. 14 is derived from the nonhomogeneous Robin boundary condition, which is inherently different from that of previous MPNN residual connections.
>
> On the theory and guarantees:
> Residual connections are primarily heuristic additions (e.g., identity mappings), whereas our $\gamma$-term arises rigorously from Riemannian boundary condition in Theorem 5.4.
> The proposed external input has theoretical guarantee. Concretely, the polynomial decay constraint for Dirichlet energy in Lemma 5.3 holds only at boundary nodes and vanishes in the interior (Remark B.6). Thus, applying external input at boundary nodes ensures the mitigation of over-smoothing.
>
> ### **4. The sentence in Line 160**
> We sincerely appreciate your comment, and will paraphrase this sentence for clarity as follows:
> “Edge deletion without careful consideration may hinder the information flow in message passing.”
>
> Indeed, we recognize the sophisticated graph rewriting techniques, and have cited [1], [2], and [4] in the Related Work as follows:
> ```
> In the line of graph rewriting, [1, 2, WWW’23] introduce edge addition-deletion algorithms in light of oversmoothing; [4] increases the spectral gap via edge deletion.
> ```
>
> Though the advanced methods of carefully rewriting the graph structure are effective, it is still challenging to study the edge addition-deletion strategy. Thus, we explore an alternative direction – refine the message passing paradigm while maintaining the original structural topology and the computational complexity of traditional MPNNs.
>
> ### **5. Parameter sensitivity on MLP**
> In GBN, we utilize a 2-layer MLP with a hidden layer dimension of 256 by default. We conduct a sensitivity analysis on the hyperparameters of the layer and the hidden layer dimension, and GBN is generally robust to these hyperparameters.
>
> |Layers|hidden_dims|CS|Ratings|
> |---|---|---|---|
> |2|128|95.84±0.23|53.34±0.52|
> |2|256|95.84±0.33|53.14±0.34|
> |2|512|95.78±0.21|53.51±0.88|
> |3|128|95.59±0.32|51.22±0.33|
> |3|256|95.65±0.11|50.98±0.51|
> |3|512|95.58±0.32|50.70±0.56|
> |4|128|95.11±0.25|49.95±0.73|
> |4|256|94.97±0.24|49.22±0.78|
> |4|512|95.07±0.11|49.73±0.38|

---

> > ### Comment · Reviewer_wr1V · 2025-08-04
> > **Thanks for the rebuttal**
> >
> > I am generally satisfied with the rebuttal—your contributions are now much clearer, and I have raised my rating to 4.
> >
> > However, the Augmented Forman–Ricci Curvature [2] is still a purely local measure, as its computation merely counts triangles and quadrangles incident on each edge and their node degrees. The authors should perform a more thorough literature review before diving into their work and use claims like “for the first time” more sparingly.

---

> > > ### Author Response · Authors · 2025-08-04
> > >
> > > Thanks again for your constructive comments!
> > >
> > > We are so glad to address your concerns and receive your appreciation! We will refine the statement and include the reference in the final version.

---

### Official Review · Reviewer_nz6f · 2025-06-28

**Clarity:** 2
**Significance:** 3
**Originality:** 3
**Rating:** 5
**Confidence:** 3

**Summary:**

The paper presents a novel local approach to tackle both oversmoothing and oversquashing in message passing neural networks (MPNNs). By integrating local Riemannian geometry, the authors design a Graph Boundary conditioned Neural network (GBN) that adaptively adjusts message passing based on local graph structures. The approach is supported by theoretical guarantees and experiments show that GBN achieves superior performance on various graph datasets.

**Questions:**

1. In the “Message of the Section” on line 162, it is stated that the method of increasing the spectral gap to reduce oversquashing impairs the effectiveness of message passing. However, in the “A more intuitive explanation” on line 158, it is mentioned that reducing oversquashing can be achieved by adding edges to allow more messages to flow. These two statements seem contradictory. I would like the authors to clarify whether the message passing becomes more unobstructed or more hindered when reducing oversquashing. A clear and consistent explanation would increase my evaluation of the paper's logical coherence.
2. Are the graph rewriting operations (adding or deleting edges) and the operations for adjusting the spectral gap independent of each other? It is not clear from the paper how these two aspects interact. If the authors can elaborate on their relationship, such as whether one operation affects the other or if they can be applied simultaneously in a coordinated manner, it will enhance my understanding of the proposed approach and potentially increase the paper's score in terms of comprehensiveness.
3. In the ablation study, when the learnable and adjustable $\gamma$ and $\beta$ are fixed to $\gamma_0$ and $\beta_0$, the accuracy does not drop significantly. I am curious about how these fixed values $\gamma_0$ and $\beta_0$ were selected. If there was a non-learning heuristic method used to choose them, please describe it. Additionally, if such a method was used, does it lead to a reduction in computational complexity? A detailed response on this would improve my assessment of the practicality and methodological soundness of the ablation study in the paper.

**Ethical Concerns:**

["NO or VERY MINOR ethics concerns only"]

**Final Justification:**

1. The comparison to the related work in your rebuttal is good. Please consider adding it to the camera-ready version.
2. "both over-squashing and over-smoothing affect the effectiveness of message-passing (MPNNs)." should be stated explicitly in your paper.
3. Thanks for clarifying the relationship between graph rewriting and adjusting the spectral gap for me.
4. Compared to determining $\gamma$ and \beta using grid search, learning $\gamma$ and $\beta$ through a learning approach indeed involves less computational effort, which is a good design.
I will keep my score.

**Limitations:**

yes

**Paper Formatting Concerns:**

According to the LaTeX style file, it is preferable that "Limitations" be set as a separate section.

**Quality:**

3

**Strengths And Weaknesses:**

The paper demonstrates notable strengths in its theoretical rigor and experimental thoroughness, with theorems (e.g., 4.2, 4.3) formally guaranteeing the approach's efficacy against oversmoothing and oversquashing, and comprehensive ablation studies validating component contributions. The experimental setup is standard and reproducible, providing clear details for experts.

However, its clarity suffers from dense theoretical notation (e.g., integral symbols and Riemannian geometry terms) that lack self-contained definitions, potentially alienating non-specialists. In significance, the work makes substantial contributions to MPNN research by addressing long-standing challenges, though its impact on broader fields remains limited. Regarding originality, while the integration of local Riemannian geometry is innovative, the paper could enhance distinctiveness by explicitly contrasting with related work (e.g., prior studies on Riemannian geometry and MPNNs cited in line 83), clarifying how the proposed GBN uniquely advances the paradigm. Strengthening these comparative analyses would solidify its claim to originality.

---

> ### Author Rebuttal · Authors · 2025-07-30
>
> ### **1. Add explicit comparison to the related work in Line 83**
>
> References 17, 34, 35 and 47 are curvature-guided graph rewriting methods, which mitigate over-squashing through adding and/or deleting edges guided by curvature. Different from this line of work altering the original graph structure, we maintain the original graph structure while mitigating over-squashing with theoretical guarantees.
>
> References 5, 10, 30, 33 and 53 extends MPNNs to certain Riemannian manifolds, following the traditional message-passing mechanism. In contrast, we propose a new, adaptive message-passing with nonhomogeneous boundary condition to jointly mitigate over-squashing and over-smoothing.
>
> ### **2. On the consistency of increasing spectral gap and adding edges w.r.t. over-squashing**
>
> Note that the topological bottleneck leads to over-squashing, but there exist other reasons resulting in over-squashing as well. (Indeed, a topological bottleneck is a sufficient but not necessary condition for causing over-squashing.)
> Also, both over-squashing and over-smoothing affect the effectiveness of message-passing (MPNNs).
> - Adding edges, which enlarges the topological bottleneck, can be beneficial for mitigating over-squashing.
> - The increase of the spectral gap $\lambda$ enlarges the topological bottleneck, as given in Theorem 4.1. However, the increase of spectral gap $\lambda$ also tends to accelerate gradient vanishing (Theorem 4.2) and over-smoothing (Theorem 4.3). Accordingly, we state that the increase of spectral gap $\lambda$ also has negative impacts on the expressiveness of MPNNs.
> The two statements are indeed consistent, and the key point here is that the topological bottleneck is not the only reason for over-squashing.
>
> Hence, over-squashing cannot be mitigated by merely increasing $\lambda$ without considering other factors, e.g., gradient vanishing. It motivates our design of local adjustments taking into account both factors.
>
> ### **3. Relationship between graph rewriting and adjusting the spectral gap**
>
> On the one hand, graph rewriting directly alters the spectral gap $\lambda$:
> As in line 138 and Theorem 4.1, the spectral gap relates to the most bottlenecked region (characterized by Cheeger's constant) in a manifold or graph. The graph rewriting expands the bottlenecks to enlarge the spectral gap.
>
> On the other hand, adjusting the spectral gap does not necessarily require graph rewriting:
> The message bottleneck is related to the graph topology and the strategy of message passing over the topology. Also, the spectral gap can be regulated without modifying the graph topology, and we leverage such fact in our work.
>
> As proved in Sec. 5, GBN achieves this by introducing boundary conditions, the adaptive Robin condition (Eq. 13).
>
> ### **4. On $\gamma$ and $\beta$**
>
> In the ablation study, the fixed values $\gamma_0$ and $\beta_0$ were determined through a grid search on the validation set, $\gamma_0 \in [0,1]$, $\beta_0 \in [0,2]$. This process required significant computational effort to identify the optimal values.
> In addition, we observe that the optimal values vary across different graphs. Graphs are diverse, and the optimal boundary conditions are inherently related to the structural patterns. Therefore, we let \gamma and \beta be learnable in GBN.

---

> ### Comment · Reviewer_nz6f · 2025-08-05
>
> 1. The comparison to the related work in your rebuttal is good. Please consider adding it to the camera-ready version.
> 2. "both over-squashing and over-smoothing affect the effectiveness of message-passing (MPNNs)." should be stated explicitly in your paper.
> 3. Thanks for clarifying the relationship between graph rewriting and adjusting the spectral gap for me.
> 4. Compared to determining $\gamma$ and \beta using grid search, learning $\gamma$ and $\beta$ through a learning approach indeed involves less computational effort, which is a good design.
> I will keep my score.

---

> > ### Author Response · Authors · 2025-08-05
> >
> > Thanks again for your appreciation! We improve the presentation accordingly.

---

### Decision · Program_Chairs · 2025-09-17

**Decision:**

Accept (poster)

**Comment:**

This paper introduces a method called GBN to overcome oversmoothing and oversquashing in graph neural networks, with theoretical underpinnings and strong empirical performance (I find the improved performance on Cora as depth increases a notable accomplishment). Reviewers are anonmyous in recommending acceptance, as do I.

I request that the authors address the concerns voiced by the reviewers in the camera ready version, mostly related to more comprehensive discussion of related work and spacing issues.